# Rapid Holocene bedrock canyon incision of Beida River, North Qilian Shan, China

Yiran Wang[1,2], Michael E. Oskin[1], Youli Li[3], Huiping Zhang[4]

[1] Department of Earth and Planetary Sciences, University of California, Davis, California, USA
[2] Earth Observatory of Singapore, Nanyang Technological University, Singapore
[3] College of Urban and Environmental Sciences, Peking University, Beijing, China
[4] Institute of Geology, China Earthquake Administration, Beijing, China

*Correspondence to*: Yiran Wang (yrwwang@ucdavis.edu)

**Abstract.** Located at the transition between monsoon and westerly dominated climate systems, major rivers draining the
western North Qilian Shan incise deep, narrow canyons into latest Quaternary foreland basin sediments of the Hexi Corridor. Field surveys and previously published geochronology show that the Beida River incised 130 m at the mountain front over the Late Pleistocene and Holocene at an average rate of 6 m/kyr. We hypothesize that a steep knickzone, with 3% slope, initiated at the mountain front and has since retreated to its present position, 10 km upstream. Additional terrace-dating suggests that this knickzone formed around the mid-Holocene, over a duration of less than 1.5 kyr, during which incision accelerated from
6 m/kyr to at least 25 m/kyr. These incision rates are much faster than the uplift rate across the North Qilian fault, which suggests a climate-related increase in discharge drove rapid incision over the Holocene and formation of the knickzone. Using the relationship between incision rates and the amount of base level drop, we show the maximum duration of knickzone formation to be ~700 yr and the minimum incision rate to be 50 m/kyr. We interpret that this period of increased river incision corresponds to a pluvial lake-filling event at the terminus of the Beida River and correlates with a wet period driven by
strengthening of the Southeast Asian Monsoon.

## 1 Introduction

An incising river responds to tectonic or climatic perturbation by adjusting its slope, expressed by formation of knickpoints or knickzones (Crosby and Whipple, 2006; Tucker and Whipple, 2002; Whittaker, 2012), and through changes of its channel width (Finnegan et al., 2005). Understanding the evolution and migration of knickzones, channel width, and the coupling
between these adjustments, is important in unravelling the type, duration, and amplitude of a perturbation (Attal et al., 2011; Berlin and Anderson, 2007; Bishop et al., 2005). Previous studies on headward migrating knickpoints focus on the role of tectonic uplift or a base level fall, and usually regard climate conditions and channel width as constant (e.g. Tucker and Whipple, 2002; Crosby and Whipple, 2006; Haviv et al., 2006; Wobus et al., 2006). Here we present a case of steep, quickly retreating knickzones within the western North Qilian Shan, formed under the combined influence of climatic change and lithologic
control. Through modelling of incision of the Beida River, as recorded by its profile and stream terraces preserved along its

course, we suggest this knickzone was formed during a short period, four to five thousand years before present, under an exceedingly fast incision rate. This is most likely to be the result of an increase in river discharge, and perhaps a commensurate decrease of sediment supply.

In western China, the North Qilian Shan is the source of several northeast flowing rivers with deep canyons incised across the mountain-basin boundary (Figure 1a). In the western North Qilian Shan, three major tributaries of the Hei He drainage: Maying, Hongshuiba, and Beida, carve deep canyons into the foreland sediments and across the fault-controlled boundary with the bedrock hinterland, forming prominent knickzones within the hinterland that are tens of meters high (Figure 1b-d). As one of these deeply incised rivers, the Beida River is characterized by a prominent knickzone which separates its profile into three patches (upper patch 1, knickzone patch 2, and lower patch 3; Figure 2a). Each patch can be distinguished by different channel slopes and channel widths: gentle and wide upper patch, steep and narrow knickzone, and a lower patch with a gentle slope similar to the upper patch, but a narrower channel. The successive generation and retreat of these patches corresponds to different boundary conditions (cf. Royden and Perron, 2013), and together record the incision history at the mountain front.

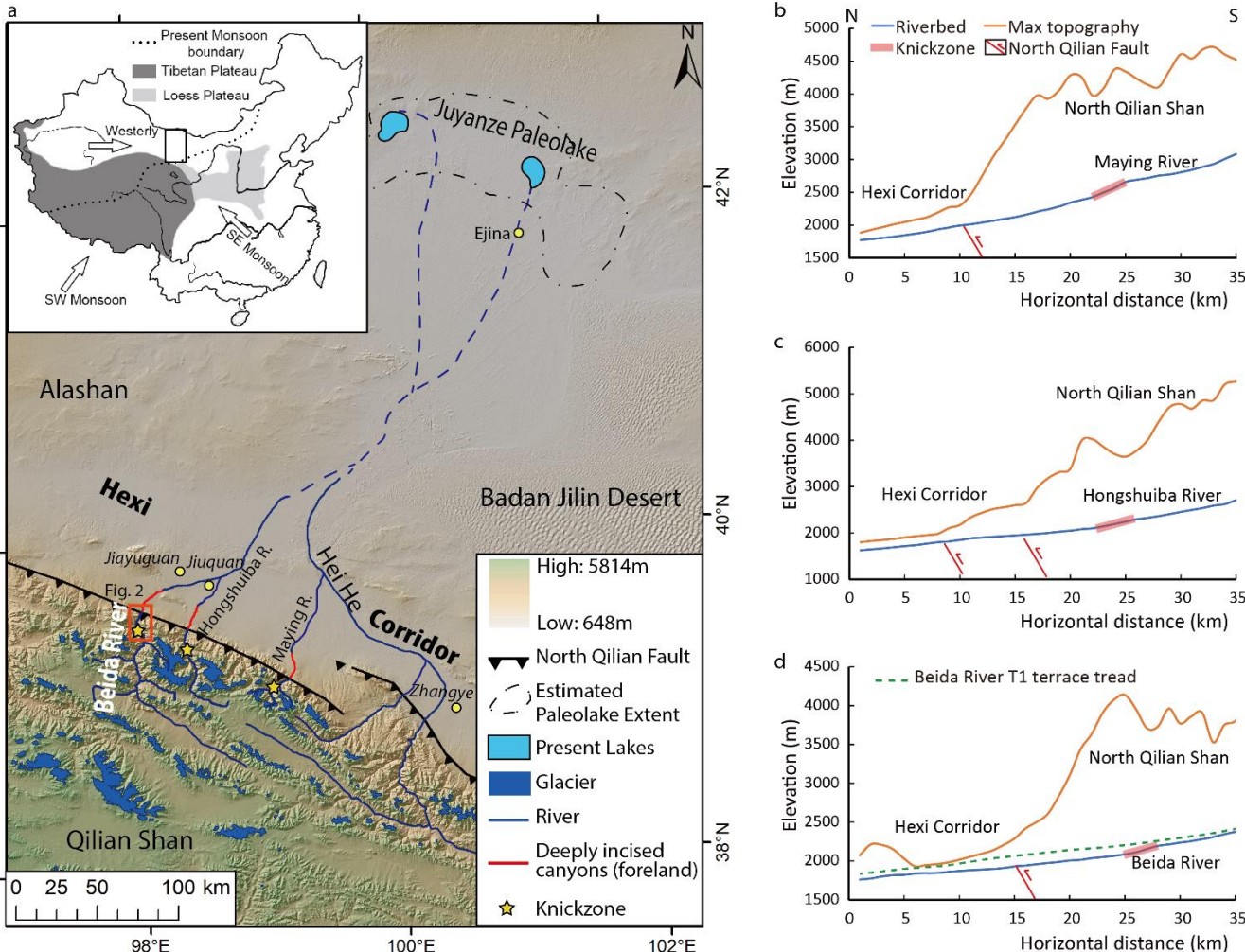

**Figure 1 a. Digital elevation map of the Hei He drainage system. Active glacial coverage map based on Raup et al., (2007). Inset figure: Location with respect to monsoon and westerly moisture sources. b-d. River and maximum elevation profiles of three major tributaries of the Hei He drainage, extracted from 30 m SRTM. Maximum elevation sampled from swaths 16 km in width, centred on each river. Locations of active strands of the North Qilian fault denote foreland-hinterland junction at mountain front. All three rivers exhibit deep incision below the top of foreland-basin sediments. Red highlights knickzone reach within bedrock. Dashed line in D is reconstructed profile of the Beida River from the ca. ~24 ka T1 terrace tread, prior to onset of rapid incision phase.**


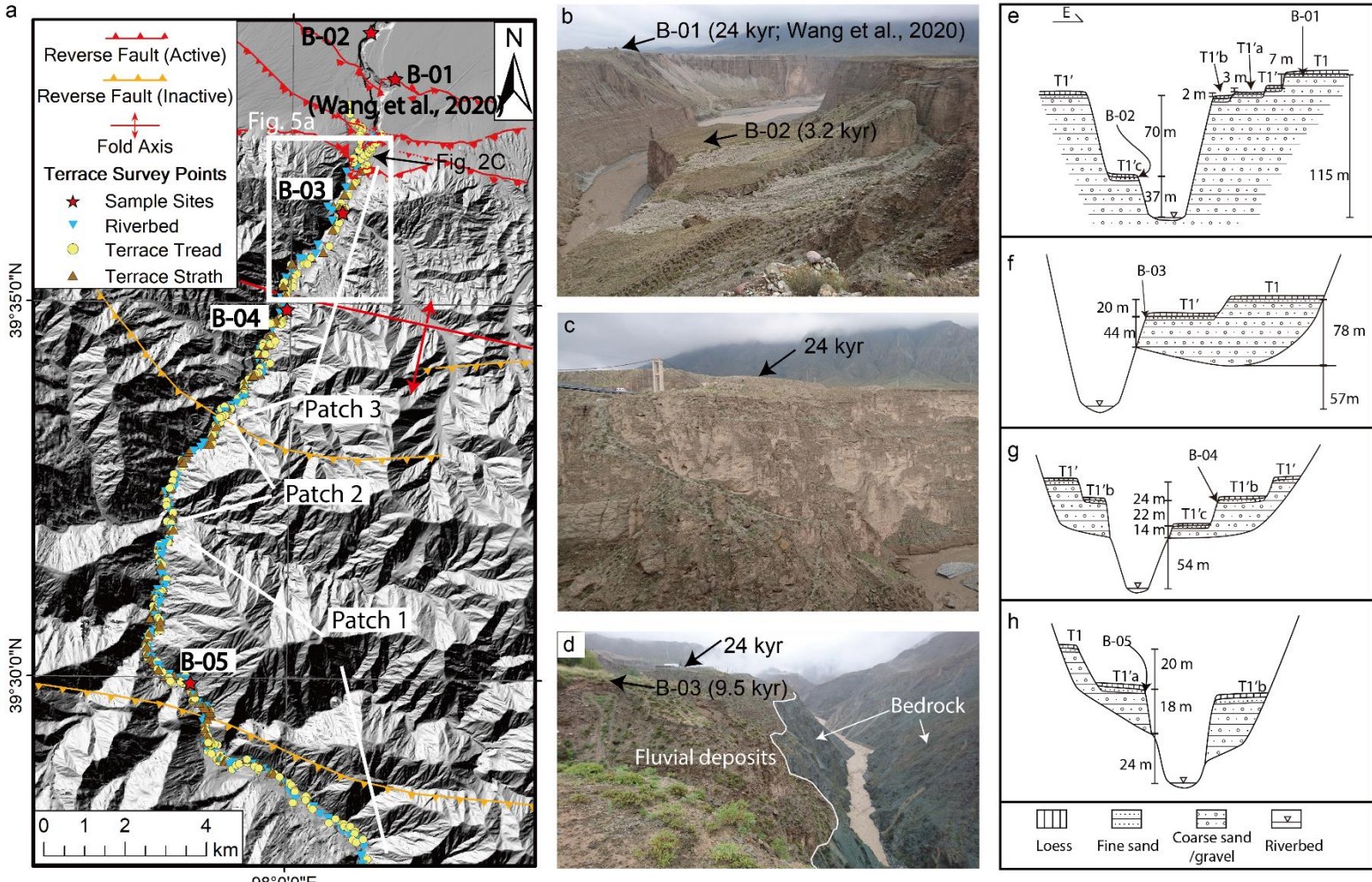

**Figure 2 a.** Map of the Beida River within the north Qilian Shan, with survey points and geochronology sample sites indicated. **b.** Photo of Beida River canyon and sample site a and b within the foreland, looking upstream. Thick layers of gravel are exposed on the canyon walls. **c.** Photo of Beida River canyon and T1 terraces near the mountain front, looking east (location in figure 2a). In this photo, the 24 kyr T1 terrace tread is ~40 m above the 9.5 kyr T1' terrace tread. **d.** Photo of Beida River canyon and sample site B-03 inside the mountains, looking upstream. This photo shows that the Beida River deeply incises into the bedrock below the terrace fill. **e-h.** cross sections of the Beida River terraces at each sample site.

## 2 Geological background

The Qilian Shan form the northeasten margin and the youngest growing portion of the Tibetan plateau (Tapponnier et al., 2001). Thermochronology studies suggest that uplift of the North Qilian Shan commenced around the mid-Miocene, with an exhumation rate of ~1 m/kyr (Jolivet et al., 2001; Zheng et al., 2010; He et al., 2017, 2021). The Hexi Corridor, north of the Qilian Shan, consists of a chain of foreland basins. Bordering arid central Asia, the Qilian Shan and Hexi Corridor occupy the transition zone between Southeast Asian Monsoon and westerlies (An et al., 2001; Wei and Gasse, 1999). The monsoon brings summer rain inland while the mid-latitude westerlies bring dry air and a small amount of water vapor in winter. Monsoon influence diminishes and annual precipitation declines from east to west, both within the Hexi Corridor and at high altitude within the Qilian Shan (Meng et al., 2012; Shi et al., 2006). At the mountain front bounding the western Hexi Corridor, the annual precipitation declines from 200-300 mm/yr at the Maying River to 100-200 mm/yr at the Beida River. At high altitude (>4000 m) within the western North Qilian Shan, there is a similar annual precipitation gradient from east to west, from 300-400 mm/yr to 200-300 mm/yr (Qiang et al., 2016; Geng et al., 2017). The maxima of precipitation occur at approximately 4200 m elevation (Chen et al., 2018). The modern glacial equilibrium line altitude of Qilian Shan increases from 4400 to 5000 m from northeast to southwest (Shi, 2011), reflecting the westward decrease in precipitation. Between the year 2005 to 2010, within the Qilian Shan there were 2684 glaciers with a total area of 1600 $km^2$ and an ice volume of 84 $km^3$ (Guo et al., 2014; Sun et al., 2015). These glaciers covered approximately 4% of the landscape above 4000 m elevation. The extent of these glaciers has fluctuated repeatedly throughout the Quaternary. Dating of moraines suggests that glacial advances have occurred during the little ice age (~1300-1850 A.D.), MIS (Marine Isotope Stage) 2, MIS 4, MIS 6, and MIS 12; some glacial expansion may have occurred during MIS 3 as well (Shi et al., 2006). The glacial equilibrium line altitude of Qilian Shan during the Last Glacial Maximum (LGM) is estimated to be as much as 400 m lower than present (Hu et al., 2014; Owen et al., 2003; Shi et al., 2006; Zhao et al., 2001; Zhou et al., 2002).

The Hei He forms the largest drainage basin in the North Qilian Shan, and terminates within the Juyanze paleolake basin, north of the Hexi Corridor. Sediment and core records from the Juyanze paleolake basin indicate frequent dry-wet oscillations over the past 11,000 yr (Hartmann and Wünnemann, 2009; Herzschuh et al., 2004; Mischke et al., 2002, 2005). The highest lake level occurred during the early-Holocene (10700-8900 yr B.P., ~20 m deep), and the highest mid-Holocene lake level (~15–17 m deep) occurred during 5400-3900 yr B.P. and peaked at about 4200 yr BP (East Juyanze lake, Hartmann and Wünnemann, 2009).

Three major tributaries, Beida, Hongshuiba, and Maying, join the Hei He from the south and west (Figure 1, table 1). In the hinterland, the three rivers flow through Pre-Cambrian and Paleozoic meta-sedimentary and meta-igneous rocks (Figure S1); in the foreland, these rivers deposit sediments into the Hexi Corridor basins. The North Qilian range overthrusts the southern margin of the Hexi Corridor, placing metasedimentary rocks against the Quaternary basin deposits. Presently, the channels of

the Maying, Hongshiba, and Beida River have incised deeply into Late Pleistocene alluvial fans of the proximal foreland basin

and into correlative fill terraces and bedrock within the range (Hetzel et al., 2019; Yang et al., 2020; Wang et al., 2020), forming a prominent knickzone along each river located ~7-13 km upstream of the mountain front (Figure 1b-d). The canyons of the three rivers are deepest at the mountain front where the North Qilian fault juxtaposes bedrock against Quaternary sediments, ~130 m, ~190 m, and ~240 m for Beida, Hongshuiba, and Maying River, respectively (Figure S2). The depths of these river canyons gradually decrease basinward until 25-30 km downstream, where the rivers form active alluvial fans.

**Table 1 hydrological information of the three major rivers draining the western North Qilian Shan (Gansu Province Local Chronicles Compilation Committee, 1998)**

| | Length (km) | Drainage area (km$^2$) | Annual discharge (x 10$^8$ m$^3$) | Glacial coverage (km$^2$); contribution to annual discharge |
|---|---|---|---|---|
| Beida | 243 | 6910 | 6.53 | 137; 16% |
| Hongshuiba | 87 | 1580 | 2.87 | 131; 35% |
| Maying | 34 | 619 | 1.16 | 20; 12% |

Along the Beida River, at least two generations of fill terraces (T1 and T2) are preserved well and continuously in the hinterland and extend to the foreland basin. Our previous research (Wang et al., 2020) suggests that T1 was abandoned after 24±3 kyr

B.P., during the last glacial period; T2 was abandoned after 144±30 kyr B.P., during the penultimate glacial period. Flights of terraces inset below both T1 and T2 terrace treads mark progressive degradation of the terrace fill and incision into underlying bedrock. Along the mountain front, strands of the North Qilian fault cut the terraces of Beida River, offsetting T1 and T2 vertically by 15 m and >60 m, respectively. In the hinterland, the T2 terrace profile reveals a long wavelength fold (~30km) with maximum uplift of ~120 m relative to T1. Combined, the fault offset and folding indicate a maximum uplift rate of ~1

m/kyr at the fold crest, and a horizontal shortening rate of 1.4±0.4 m/kyr across the North Qilian Shan (Wang et al., 2020).

## 3 Methods

### 3.1 Field survey

The late Pleistocene T1 fill terrace, up to 60-80 m thick, is preserved continuously along the narrow bedrock gorge of the Beida River. This terrace grades to an extensive alluvial fan deposit emanating from the mountain front, with minor disruption

from reverse fault offsets (Figure 2). The Beida river gorge cuts across the fault-controlled basin boundary, forming a narrow slot canyon up to 125 m deep within the foreland-basin fan gravels. We mapped and surveyed the terrace sequence and the course of Beida River inside the mountain range using a laser rangefinder (~0.3 m distance accuracy, 0.25° inclination accuracy) and differential GPS. Wherever possible, the terrace tread (top of the fluvial gravel), terrace strath (base of fluvial gravel), and present riverbed were measured together (Figure 3, Table S1). In the foreland, we extract terrace elevations and the river profile

from an 8 m resolution digital elevation model produced by the Polar Geospatial Center (Shean, 2017). For the hinterland

tributaries, we extract river profiles from the same 8 m resolution DEM. Present bedrock channel widths were measured from Google Earth imagery at 100 m intervals along the river course. We measured the width of the water surface from imagery acquired during the wet season, mostly between July to September, 2010 to 2016. Due to limited data availability, a few measurements were obtained from imagery acquired in May and October (Table S2).

**3.2 Geochronology**

The abandonment age of T1 was dated to be 24±3 kyr by combining optically stimulated luminescence (OSL) and Terrestrial Cosmogenic Nuclide ([10]Be) exposure ages (Figure 2, sample site B-01; Wang et al., 2020). To document the post-24 kyr incision history of Beida River within the hinterland, we collected charcoal samples from the fine sand and silty overbank deposits on three inset terraces (Figure 2, sample site B-03, B-04, B05, Figure S3). These overbank deposits were deposited

after terrace formation, but before incision was sufficient to isolate the terrace surface from flood events. At one site within the foreland basin, we collected an OSL sample from the bottom of the loess covering a low inset terrace (Figure 2, sample site B-02). Ten charcoal samples were measured at the Keck Carbon Cycle AMS Facility at UC Irvine. The results were calibrated with IntCal13 calibration curve (Reimer et al., 2013) (Table 2). The OSL sample (BD-O-12) was processed and measured at the State Key Laboratory of Earthquake Dynamics, China Earthquake Administration. The equivalent doses (De)

for the pure fine-grained quartz were determined by the simplified multiple aliquot regenerative-dose (SMAR) protocol (Table 3, Table S3, Figure S4).

**Table 2 [14]C age of Beida River terraces**

| Sample site[1] | Coordinates | Sample name | fraction Modern | ± | D[14]C (‰) | ± | [14]C age (BP) | ± | Calibrated age 1σ (BP) | 2σ (BP) |
|---|---|---|---|---|---|---|---|---|---|---|
| **B-03** (T1'; 115 m) | 39.60126°, 98.01365° | BDC-3 | 0.3462 | 0.0028 | -653.8 | 2.8 | 8520 | 70 | 9473-9544 | 9332-9340<br>9404-9632<br>9645-9657 |
| | | BDC-4 | 0.3393 | 0.0008 | -660.7 | 0.8 | 8680 | 20 | 9557-9630<br>9647-9653 | 9552-9679 |
| | | BDC-5 | 0.3378 | 0.0008 | -662.2 | 0.8 | 8720 | 20 | 9611-9699 | 9561-9737 |
| | | BDC-6 | 0.3320 | 0.0009 | -668.0 | 0.9 | 8855 | 25 | 9895-9949<br>9990-10012<br>10025-10038<br>10061-10134 | 9784-9848<br>9861-9878<br>9883-9966<br>9982-10155 |

| Sample Site | Coordinates | Sample name | | | | | | | | |
|---|---|---|---|---|---|---|---|---|---|---|
| | | BDC-8 | 0.5973 | 0.0011 | -402.7 | 1.1 | 4140 | 15 | 4617-4652 4669-4703 4757-4765 4784-4809 | 4580-4726 4752-4770 4780-4815 |
| B-04 (T1'b; 90 m) | 39.57600°, 97.99493° | BDC-9 | 0.5715 | 0.0011 | -428.5 | 1.1 | 4495 | 20 | 5054-5077 5105-5136 5163-5189 5213-5228 5231-5251 5257-5281 | 5047-5147 5153-5202 5210-5288 |
| | | BDC-10 | 0.5862 | 0.0011 | -413.8 | 1.1 | 4290 | 15 | 4844-4856 | 4839-4862 |
| | | BDC-11 | 0.4702 | 0.0012 | -529.8 | 1.2 | 6060 | 20 | 6893-6944 | 6807-6811 6856-6979 |
| B-05 (T1'a; 42 m) | 39.498771°, 97.971940° | BDC-12 | 0.4497 | 0.0029 | -550.3 | 2.9 | 6420 | 60 | 7309-7419 | 7185-7186 7246-7439 |
| | | BDC-14 | 0.4725 | 0.0010 | -527.5 | 1.0 | 6025 | 20 | 6800-6815 6845-6901 | 6797-6934 |

1. Inside the bracket shows the terrace that the sample collected from, and the height of the terrace above the riverbed. Note that labels a, b, and indicate local terrace ordering, and cannot be correlated to terraces upstream and downstream.

**Table 3 OSL age of loess covering terrace tread**

| Sample Site | Sample name[1] | Coordinates | U /ppm | Th /ppm | K (%) | Water Content (%) | Dose Rate (Gy/ka) | Equivalent Dose [1] (Gy) | Age [2] (ka) |
|---|---|---|---|---|---|---|---|---|---|
| B-02 | BD-O-12 | 98.02299, 39.64376 | 2.34±0.10 | 9.32±0.28 | 1.67±0.06 | 0 | 3.5±0.3 | 11.4±0.7 | 3.2±0.2 |

1. Grain size <100 μm

2. Uncertainties in equivalent dose, dose rate and age determinations are expressed at the 1σ confidence level.

**3.3 Bedrock incision model**

We apply the concept of slope patches (Royden & Perron, 2013) to model the evolution of the Beida River stream profile. The formation of a slope patch is based on stream power, which has the form

$$\frac{dz}{dt} = K \left(\frac{QS}{W}\right)^{n} \quad , \quad (1)$$

where z is the channel elevation, t is time, Q is river discharge, S is channel slope, W is channel width, and K and n are an empirical erosional efficiency and exponent, respectively (Tucker & Whipple, 2002; Whipple & Tucker, 1999).

A slope patch forms in the bedrock channel immediately upstream of the channel outlet, with channel slope that develops in balance with the rate of base-level fall. Setting $\frac{dz}{dt}$ to the incision rate, $I$, at the mountain front, we rearrange equation 1 to solve for this channel slope:

$$S = \left|\frac{dz}{dx}\right| = \left(\frac{I}{K}\right)^{\frac{1}{n}}\left(\frac{W}{Q}\right) \qquad (2)$$

For the case of the Beida River, no major tributary enters along its lower 30 km long course; the drainage area of the Beida River, measured from 30 m SRTM DEM, at the river outlet (mountain front) is $\sim 6.91 \times 10^9$ km$^2$, while the drainage area above the knickzone is $\sim 6.84 \times 10^9$ km$^2$, a difference of $\sim 1\%$. Thus, we assume that Q does not vary spatially along the channel course and may be treated as constant over an incision phase, though it may vary from one phase to the next. We also assume that channel width (W) remains constant within a given slope patch. During formation of a slope patch, river profile elevation is thus found by integrating equation 2 over its finite span $x_b$ to x:

$$z(x) = \left(\frac{I}{K}\right)^{\frac{1}{n}}\left(\frac{W}{Q}\right)(x - x_b) + z_b = S(x - x_b) + z_b \qquad (3)$$

where $x_b$ and $z_b$ are the horizontal position and elevation at base level, respectively. We define base level as the bedrock-basin transition at the mountain front.

We model the bedrock incision history of the Beida river as a consequence of varying incision rate over time at base level. Once formed, a slope patch behaves as a kinematic wave, retaining its gradient as it retreats upstream (Perron and Royden, 2013). The elevation of the (n-1)th slope patch (the patch formed one stage before present) may thus be cast as a function of its slope during formation, $S_{(n-1)}$ and an effective base-level elevation $z_{b(n-1)}$ of the slope patch projected to the outlet position. This base level may be predicted by correcting the present base level elevation, $z_b$, by the difference in the amount of incision across neighboring patches n and n-1,

$$z_{b(n-1)} = z_b + \left(I_{n,j} - I_{n-1,j}\right)t_j. \qquad (4)$$

$I_{n,j}$ is the incision rate of patch n, currently being formed during time interval $t_j$, directly upstream of the outlet. $I_{n-1,j}$ is the incision rate of patch n-1 during that time interval $t_j$. Note that this incision rate may be different than the incision rate during formation of patch n-1 (i.e. faster for an increase in discharge). However, varying discharge over time is not required for our model, only changes of channel slope driven by varying of incision rate at the mountain front. For the (n-2)th patch, the effective base level contains two correction terms (see $z_{b1}$, $z_{b2}$ of figure 3c),

$$z_{b(n-2)} = z_b + \left(I_{n,j} - I_{n-2,j}\right)t_j + \left(I_{n-1,j-1} - I_{n-2,j-1}\right)t_{j-1}. \qquad (5)$$

This may be generalized to additional slope patches, each corrected by the incision rate differences between patches. We apply equations 4 and 5, combined with the incision recorded in stream terraces adjacent to the Beida River, to constrain its incision-

rate history. Note that eqs. 4 and 5 are derived on the premise that discharge (Q) is constant during each incision phase, and that channel width (W) may vary between different patches but remains constant within a patch – i.e., that eq. 3 is valid.

## 4 Results

### 4.1 Beida River profiles

Presently, the 30 km reach of the Beida River upstream of the mountain front is entirely contained within a bedrock channel. Channel slopes, measured directly from fitting the long profile, show a knickzone between 10 to 12 km upstream of the mountain front (Figure 3). The knickzone divides the river profile into three patches: patch 1, upstream of the knickzone, with slope of 0.013; patch 2, the knickzone itself, with slope of 0.029; patch 3, below the knickzone with slope of 0.012 (Figure 3c). Channel width also varies along the course of the Beida River. In patch 1, the channel width ranges between 14 to 140 m, with an average of 43 m; in patch 2, the channel width ranges between 8 to 29 m, with an average of 17 m; in patch 3, the channel width ranges between 14 to 70 m, with an average of 30 m (Figure 3d). In the foreland, the river incises into the 24 kyr T1 terrace, which is part of an extensive alluvial fan surface. The river canyon is deepest at the mountain front, and its depth gradually decreases downstream, indicating a decrease of river gradient within the foreland from 24 kyr B.P. to present. Near the mountain front, the slope of the riverbed is presently ~0.010, while the slope of the adjacent alluvial fan surface is ~0.013.

Major tributaries of the Beida River all have knickpoints at their junction with the main stem (Figure 4). Below the main-stem knickzone, these tributary knickpoints are mostly over 100 m high and form waterfalls. Above the main stem knickzone, the tributary knickzones are steep but graded to the main stem, and only half as high as those downstream of the main-stem knickzone. By projecting the tributary profiles to the main stem, we find that the tributary junctions define a single, graded main-stem profile without evidence for a knickzone (Figure 4).

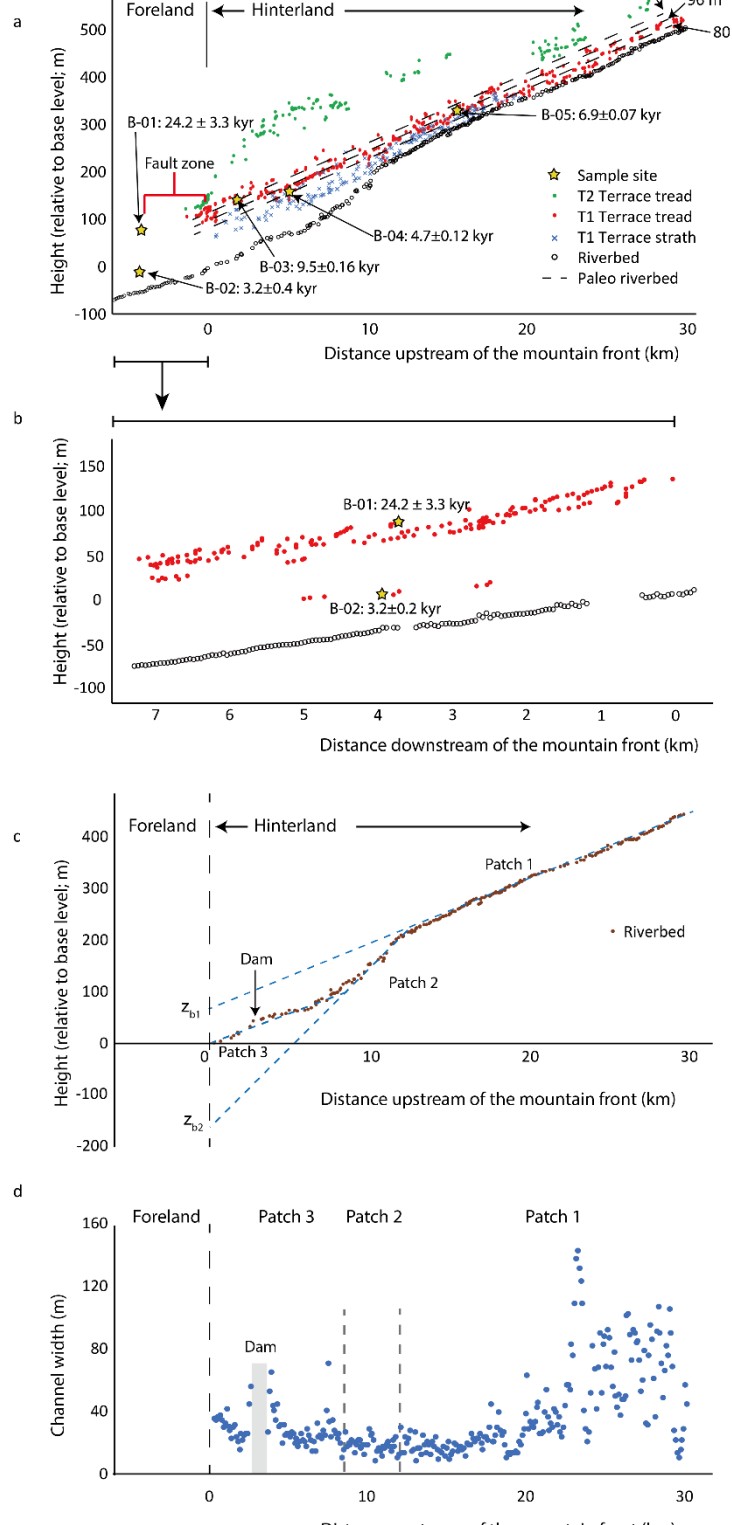

**Figure 3 a)** Longitudinal profile of the Beida River channel, with terrace treads (T2, T1 and all T1 inset terrace treads) and strath elevations; sample sites and ages (expressed at the 2σ confidence level) also indicated. Dash lines are inferred prior river profiles corresponding to the sample site elevations, with the relative height above present base level annotated (not corrected for tectonic uplift). **b)** terrace elevations (red) and channel elevations (open circles) along the Beida River within the foreland basin. Note sparse terrace record below T1/T1'. **c)** Three patches of Beida river profile, projected to effective base level at the mountain front. **d)** Present bedrock channel width measured at 100m intervals along the course of the Beida River. The width measurements that are within ~500 m upstream & downstream of the dam are excluded.

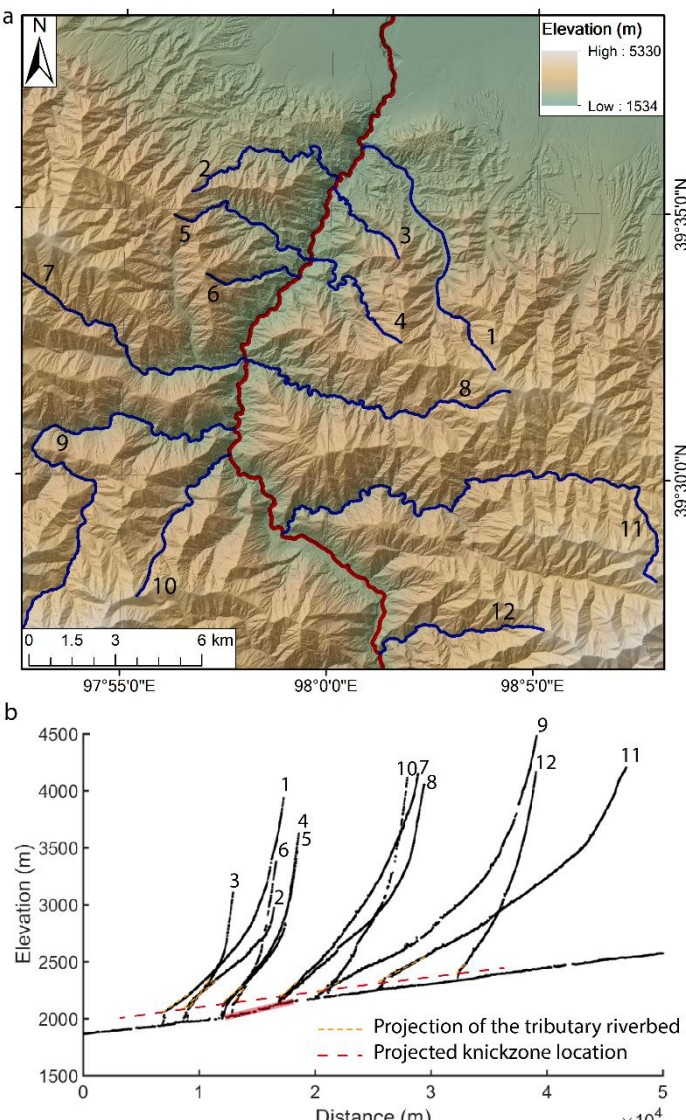

**Figure 4 a.** Map of the tributaries of the Beida River. Red delineates the main stem, blue delineates tributaries. **b.** Longitudinal profiles of the tributaries of the Beida River, as numbered in a. The main-stem knickzone is highlighted in red. Dashed red line connects the projection of tributary streams to the junction with the main stem based on stream gradient above each tributary knickzone.

## 4.2 Beida River terrace development

North of the mountain front, the 24 kyr T1 fill terrace merges into the extensive T1 alluvial fan. Though the river has incised an over 100 m deep canyon, no bedrock is exposed in the foreland. Within the mountain range, the T1 fill ranges from 60 to 80 m of thickness, and consists of unconsolidated medium to poorly sorted, well-rounded boulder-cobble conglomerate and sandy conglomerate. The lithology of the sediments mainly consists of quartzite, granite, slate, and limestone. T1 treads are very well preserved, only covered by 1-2 m loess cap except at tributary junctions, where alluvial fans are deposited upon the tread. The T1 tread presently lies ~115 m above the riverbed immediately north of the northernmost strand of the North Qilian fault, ~125-130 m above the present riverbed of patch 3 at the mountain front, up to ~130-135 m above the patch 3 upstream of the mountain front, and ~60 m above the riverbed of patch 1. Bedrock below the T1 strath is exposed continuously in the hinterland for at least 30 km along the Beida River. The most prominent inset (cut) terrace, T1', is preserved continuously at an elevation 7 to 20 m lower than T1 tread, both inside the mountain range and in the foreland basin. [14]C dating of charcoal samples at site B-03 (Figure 2a, 2f, 3a) indicates abandonment of T1' occurred after 9.5 ± 0.16 kyr BP. Several levels of local inset (cut) terraces are preserved along the river, generally between a few meters to ~40 m below the T1' tread. Different from T1 and T1' terraces, these inset terraces are local features formed as the river incised and meandered (Merritts et al., 1994) and therefore it is difficult or impossible to correlate these to other terraces upstream and downstream. All inset terraces are cut into the T1 fill. No inset terraces appear to have been formed as the river cut below the level of the T1 fill. On top of all the inset terrace treads is half to 1 m thick fine sand layers with small gravel (<10 mm) layers in between, which we interpret as overbank sediments deposited during floods soon after the terrace surface was abandoned. Loess, 0.5 to 1.5 m thick, is deposited atop the overbank deposits.

Selected inset terraces were dated to constrain the incision history of the Beida River. An inset terrace at site B-05, situated 42 m above the present riverbed of patch 1, yields an age of 6.9 ± 0.07 kyr (Figure 2a, 2h, 3a, Table 2). At site B-04, an inset terrace with tread elevated 90 m above the patch 3 riverbed and ~24 m below the T1' tread yields an age of 4.7 ± 0.12 kyr (Figure 2a, 2g, 3a, Table 2). In the foreland, a suite of terraces was formed as the river first incised into the basin deposits. Here T1' and other younger inset terraces occur between 7 and 30 m below the T1 tread and 80 to 100 m above present river level. Terraces are absent between 80 m and ~30-40 m above the riverbed (Figure 3a, 3b). An OSL sample collected from basal loess capping an inset terrace tread, 37 m above the present riverbed, yielded an age of 3.2 ± 0.40 kyr (site B-02 of figure 2a, 2e, 3a; Table 3).

We find that the Beida River incised its present bedrock gorge adjacent to its former canyon filled by T1. This appears to have started with formation of the T1' terrace at 9.5 kyr. Upstream of the mountain front, the T1' terrace tread is preserved continuously on the east bank for 3.6 km, while only bedrock is exposed on the west bank. Though the T1 terrace fill is up to ~60-80 m thick in this area, bedrock is exposed in the canyon walls 10 to 15 m below T1' and 20 to 30 m below the top of the T1 fill. Farther upstream, mapped exposures show that the entire paleo-channel fill has been preserved beneath T1 and T1' for several channel reaches (Figure 5, S5).

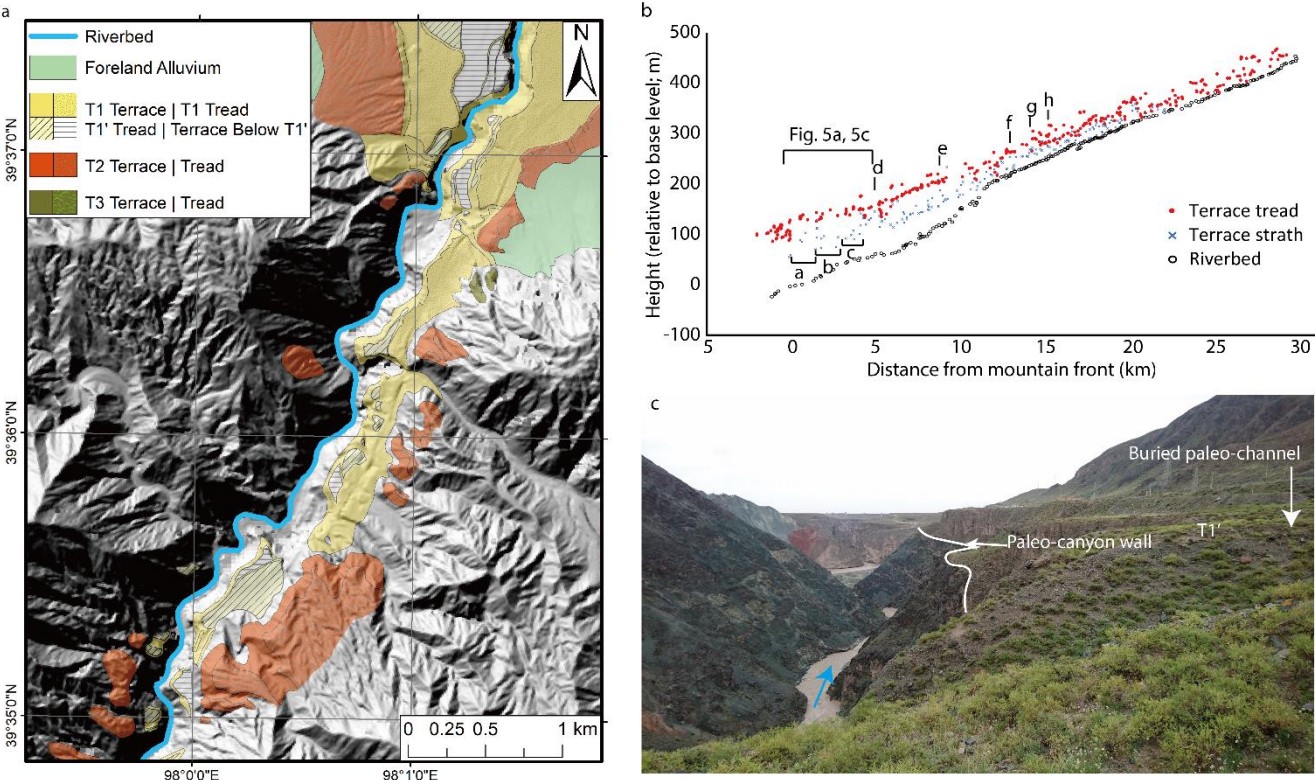

**Figure 5 Bedrock incision and terrace preservation along the Beida River. a. Terrace map of the lower reach of Beida River. T1 terraces are all preserved on the east side of the canyon. b. Terrace profiles of Beida River. Letters correspond to reaches where terrace mapping indicates that the original paleocanyon is preserved and the river incised into the adjacent bedrock soon after the abandonment of T1' (See Figure S5). c. Photo of the lower reach of Beida River, looking downstream, with paleochannel axis and paleo-canyon wall annotated. Here the river incised mostly into bedrock along the west edge of the T1 terrace fill deposit.**

### 4.3 Incision rate estimation with terrace records

The incision rate at the mountain front may be calculated from the ages and relative heights of the T1 inset terraces. Because the sample sites are scattered along the river, the heights of these terraces cannot be compared directly. Instead, we reconstruct the elevation of the river channel at 9.5, 6.9, 4.7 kyr B.P., by projecting the channel profile through the three sample sites to the mountain front based on the slope of patch 1 (Figure 3a). The heights of these paleo channels above the present riverbed elevation at the mountain front are 110 m, 96 m, 80 m, respectively (Table 4). After correction for faulting at the mountain front and folding of the range interior (Wang et al. 2020), the elevations at the mountain front are 102 m, 89 m, and 75 m above the present riverbed, respectively (Table 4). For the foreland terraces, we project the 3.2 kyr terrace to the mountain front based on present alluvial channel slope, retaining an elevation of 37 m above the river. Because this terrace is located on the footwall of the North Qilian Fault, we do not apply an adjustment for tectonic uplift. We also do not correct this site for subsidence, because unlike for the hinterland, where we have good constraints on uplift from deformed terraces, we do not have a marker of subsidence of the foreland. Wang et al. 2020 estimated that tectonic subsidence is only a small fraction (<10%) of the

hangingwall uplift and thus such a correction would be less than 0.5 m for this terrace. For the 24 kyr T1 tread, we use the T1 height at the mountain front, 131 m. After adjusting for tectonic uplift (15 m), its corrected elevation is 116 m. With these adjusted terrace heights, we calculate the incision rate at mountain front between 24.2 to 9.5 kyr B.P. was ~1 m/kyr. The

incision rate accelerated over much of the Holocene, from ~5 m/kyr between 9.5 to 6.9 kyr B.P., to ~6 m/kyr between 6.9 to 4.7 kyr B.P., and to ~25 m/kyr, between 4.7 to 3.2 kyr B. P. The incision rate remains high, at ~12 m/kyr, from 3.2 kyr B. P. to present (Figure 6). It is worth noting the 3.2 kyr terrace date is a minimum age, which may lead to overestimation of the post-3.2 kyr incision rate and underestimation of incision rate between 4.7 and 3.2 kyr. However, we consider the 3.2 kyr date to be a good estimate of the terrace age because loess has been deposited continuously in this area since at least the mid-early

Holocene (Küster et al., 2006). We also note that samples from the loess cover atop of T1 terrace tread at site B-01 yield ages of 5.7 kyr and 6.5 kyr (Wang et al., 2020), which further supports that loess deposition began at this time.

**Table 4 Terrace ages and relative heights.**

| Terrace Age[1] (kyr) | Original terrace height (projected to mountain front, m) | Tectonic uplift rate[2] (m/kyr) | Terrace Height (Adjusted based on tectonic uplift) | |
|---|---|---|---|---|
| | | | Relative to present riverbed at mountain front (m) | Relative to patch 1 (m) |
| 24.2 ± 3.3 | 131 ± 3[3] | 0.62 ± 0.08 | 116 ± 3 | 64 ± 3 |
| 9.5 ± 0.16 | 110 ± 2.7 | 0.86 ± 0.24 | 102 ± 5 | 50 ± 5 |
| 6.9 ± 0.07 | 96 ± 3.2 | 0.92 ± 0.25 | 89 ± 4.9 | 37 ± 5 |
| 4.7 ± 0.12 | 80 ± 2.4 | 1.08 ± 0.29 | 75 ± 3.8 | 23 ± 3.8 |
| 3.2 ± 0.4 | 37 ± 3 | 0 | 37 ± 3 | - |

[1] terrace ages are expressed at the 2σ confidence level

[2] Tectonic uplift rates at the sample site are calculated based on the folding and faulting data in Wang et al., (2020).

[3] Here we report the terrace height at mountain front instead of the projected height from the sample site.

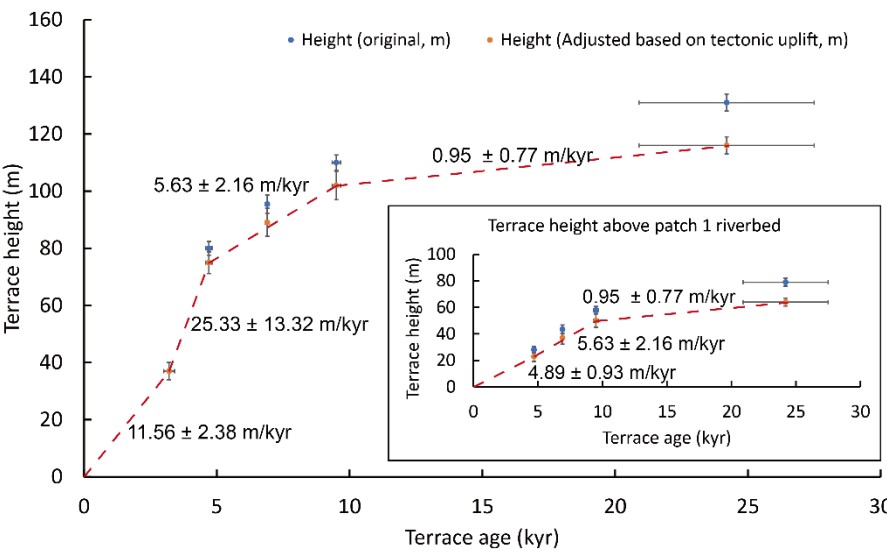

Figure 6 a) Terrace height above base level and the incision rate of each stage at the mountain front. b) Terrace height and incision rate of each stage above patch 1, upstream of the knickpoint. Details of elevations with correction for tectonic uplift are in Table 4.

## 5 Discussion

### 5.1 Channel width and bedrock incision rate

Drainage area, and thus discharge, does not appreciably change across our research area, however the bedrock channel width varies widely, from over 100 m (patch 1) to as low as ~10 m (patch 2). Based on bedrock channel erosional mechanisms, the channel width should scale with discharge and channel slope (Finnegan et al., 2005, 2007; Lamb et al., 2015; Turowski et al., 2007; Wobus et al., 2006; Wohl and David, 2008). Though the narrowing of patch 2, and a clear, almost factor of two increase in the average width from patch 2 to patch 3, are both consistent with coupling of channel width and slope, it is obvious that the upper reach of the patch 1 is considerably wider than expected (Figure 3d). In addition, the cause for narrowing of channel width on patch 1 over the 5 km reach upstream of the knickzone, is also not clear. Increased flow velocity and shear stress immediately above the knickzone (Haviv et al., 2006) could contribute to this narrowing, though it is unlikely that such a hydraulic effect would extend for kilometers upstream. Another possibility is that the wider reach of the Beida River is a result of lateral incision and removal of a more extensive T1 terrace fill. Though bedrock is exposed in the channel throughout patch 1, T1 terraces are more extensively preserved along its widest reach.

Variations in channel width may exert as much influence as discharge on incision rate (Lave and Avuoac, 2000). For the Beida River, we observe that patch 1 and patch 3 share similar channel slopes (Figure 3c), but show a two-fold difference in their

incision rates (Figure 6). We attribute this difference to the difference in average channel width between different patches (Figure 3d). We suggest that the narrower average width of patch 3 focuses stream power and enhances incision rate relative to patch 1 (Eq. 1). This explanation is at odds, however, with the 5 km long narrower reach of patch 1, which by this reasoning should be incising as fast as patch 3. We speculate that this reach could reflect recent removal of the T1 terrace and focusing of the channel as it incise into bedrock. We note that the channel slope here is slightly steeper than upstream, which could

indicate acceleration of incision rate. At present the terrace record is insufficient to test this hypothesis.

### 5.2 Beida River knickzone formation and incision stages.

We interpret the knickzone of the Beida River as a migrating kinematic wave, related to an accelerated incision rate earlier in time and downstream of its present position. We suggest the Beida River knickzone formed in response to an abrupt change in the rate of base-level lowering (Whipple and Tucker, 1999). It is likely that the knickzone of the Beida River first formed

at the mountain front, where bedrock is juxtaposed against Quaternary foreland-basin sediments by the North Qilian fault. The knickzone is hypothesized to have formed here as a response to increased incision rate into the foreland-basin sediments, thus lowering base level relative to the bedrock. The most likely cause for this increase in incision rate is a change in discharge during this time period, with perhaps a contribution from decreased sediment flux. Both mechanisms would increase transport capacity and promote incision of the unconsolidated foreland-basin sediments (Tucker and Whipple, 2002). We suggest a

change of discharge (and/or decrease of sediment flux) and the change in erodibility from foreland sediments to hinterland bedrock are the two key factors contributing to knickzone formation here. Without a change of erodibility, an increase of discharge would lead to uniform incision (if discharge is spatially invariant) of the bedrock channel, and relaxation (lowering of the river gradient) for the alluvial channel. We hypothesize instead that exposure of bedrock within the channel at the fault contact with the foreland led incision of the unconsolidated foreland-basin sediments to outpace bedrock incision upstream,

resulting in formation of the knickzone.

We can rule out the alternative hypothesis of tectonic uplift as a driver of the formation of the prominent knickzone (patch 2), for two reasons. First, judging by the continuity of the T1 terrace tread (Figure 3a) and our field work (Wang et al., 2020; figure S6), we can rule out the presence of an active fault under the knickzone. Second, the rate of incision of the Beida River greatly exceeds the tectonic uplift rate. Based on our previous research (Wang et al., 2020), the average vertical uplift rate at

the mountain front since the abandonment of T1 is only ~0.6 m/kyr. Additional uplift occurs via folding within the hinterland, reaching a maximum of ~15 m for the T1 terrace (vertical displacement by the frontal fault excluded) at 5 to 10 km upstream of the range front. This folding contributes an additional ~0.6 m/kyr to the uplift rate. In addition, the long-term exhumation rate based on thermochronology is ~1 m/kyr (Zheng et al., 2010). In comparison, the average incision rate since the abandonment of T1 terrace (24.2 kyr) is ~5.4 m/kyr, and the average incision rate since the abandonment of T1' (9.5 kyr) is

~12 m/kyr, both much larger than the tectonic uplift or exhumation rate.

A second alternative hypothesis is that the Beida River knickzone formed at its present location, related either to a change in erodibility of bedrock or a major tributary confluence. Based on a drainage-area analysis, no major tributary confluence occurs at the knickzone location. The geologic map (Figure S1) shows no significant lithology change under the present knickzone location, which suggests local variations of lithology are unlikely to have caused knickzone formation at its present position.

In addition, if the knickzone of the Beida River were a persistent feature of the channel, evidence for a buried knickzone should be preserved in the strath (paleo river bed) elevations beneath the T1 terrace. We find that T1 strath elevations show no such sign of a knickzone (Figure 3a). Therefore, neither the regional context nor local incision history suggest the presence of fixed knickzone, and this alternative hypothesis can also be ruled out.

Based on the calculated incision rates, we can divide the Beida River incision history since the abandonment of T1 into two major phases. The first phase is between 24 to 9.5 kyr B.P. with an average incision rate ~1 m/yr after correcting for tectonic uplift, representing a close to equilibrium state relative to the long-term uplift and exhumation rate of the North Qilian Shan. The second phase is between 9.5 kyr B.P. to present, with an average incision rate an order of magnitude larger than the previous phase, representing an accelerated incision state. During the first phase, the relatively slow incision rate allowed the Beida River to meander widely, not only in the foreland but also within its hinterland canyon. This meandering resulted in widening of the canyon as the river laterally eroded into the bedrock. This more or less stable state ended with the abandonment of the prominent T1' terrace. During the second phase, incision rate increased in both the foreland and the hinterland, and has been dominated by down-cutting instead of lateral erosion. Due to this change in incision rate, only local inset terraces were formed during this phase. It is also during this phase that tributary knickzones started to form in response to the increasing incision rate of the main stem. We also suggest that channel incision into bedrock began during this second phase, based on the following evidence. First, bedrock outcrops occur as high as 10 meters below the mapped remnants of the 9.5 kyr T1' tread. Second, mapping of the T1 terrace fill shows that prior to formation of T1', the Beida River had cut laterally into bedrock along the western side of its canyon over the 3.6 km reach immediately upstream of the mountain front (Figure 5). Post-T1' incision occurred in this area, preserving the T1 paleo channel axis and much of its buried canyon wall below the T1 and T1' terraces. The post-9.5 kyr B.P. increase of incision rate thus led to isolation of the river course within the bedrock below T1'.

Based on the slope patch theory for bedrock incision, we associate the three slope patches along the bedrock channel of the Beida River as formed during three incision stages over the second phase, since 9.5 kyr B.P.. In our model, each patch is projected to the mountain front outlet where the bedrock channel transitions to an alluvial channel. During the 1st stage, relatively slow incision rate in the foreland formed the gentle slope and wide channel of patch 1. During the 2nd stage, incision rate increased drastically in the foreland, which led to the formation of a steeper, narrower patch 2. During the 3rd stage, incision rate in the foreland decreased, forming the youngest patch 3 with gentle slope and wider channel, while the steeper patch 2 retreated upstream, replacing patch 1 (Figure 7). Combined with the terrace records, we therefore define the 1st incision stage to occur from 9.5 kyr to sometime around 4.7 kyr B. P., because the incision rates between 9.5 to 6.9 kyr B.P. and 6.9 to 4.7 kyr B.P. are identical within error, between ~5-6 m/kyr (Figure 6). This is followed by the 2nd stage, with an incision rate of at

least ~25m/kyr. The 3<sup>rd</sup>, present stage, started at or before 3.2 kyr B.P. with an incision rate of ~12 m/kyr. Because the starting time and the duration of stage 2 is not well constrained due to the sparse terrace record, it is possible that stage 2 has a shorter duration than 1.5 kyr and its ~25 m/kyr incision rate should be considered a minimum.

In addition to our terrace chronology, a three-stage incision history is also supported by the relative preservation of terraces along the Beida River, with a notable absence of terraces preserved between 40 and 80 m above river level (75 and 30 m below T1, figure 3b) within the foreland, during the period of most rapid incision inferred from the bracketing terrace record. These three incision stages have also left their imprints into the tributary knickzones (Figure 4). Steep knickzones, often ending at waterfalls, occur on tributaries below the main-stem knickzone, suggesting that when these tributary knickzones formed the main stem was incising very fast so that the incision rate of the tributaries could not keep pace. Gentler knickzones formed on tributaries that join upstream of the main-stem knickzone suggest slower incision rate of the main stem here, as the main-stem knickzone has not yet retreated through this area. These tributary knickzones continue to grow in height and upstream extent, as the incision rate in the main stem is still fast at present.

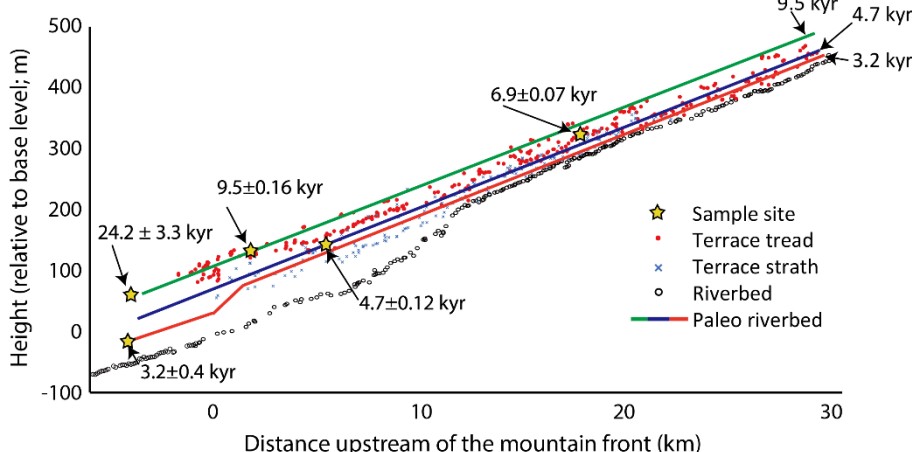

**Figure 7 Simplified model for Beida River incision since 9.5 kyr. The surveyed terrace treads and the paleo riverbeds we projected are corrected for tectonic uplift, therefore slightly deviated from the sample locations. The time period between 24.2 to 9.5 kyr B.P. is not included in this model because the river was mostly incising into the T1 terrace fill, and therefore the incision of the foreland and hinterland should be synchronous.**

### 5.3 Coupled incision model for knickzone formation

Geometric and timing relations for patches 1, 2, and 3 may be coupled to further constrain the duration and incision rate of the knickzone formation stage (stage 2). Here we formulate these relationships into two sets of constraints. Constraint 1 comes from the knickzone retreat process and the geometric relationship between patch 2 and 3. The effective base level of patch 2 ($z_{b2}$, Figure 3c) is determined by the relative incision rate between patch 2 and patch 3, and the duration of the 3<sup>rd</sup> stage. Based on equation 4, we have

$$z_{b2} = (I_3 - I_{2,3}) \times t_3 \qquad (6a)$$

Because we may estimate $I_3$ and $t_3$ from the position of the youngest, 3.2 kyr B.P. terrace, and $z_{b2}$ from projection of the patch 2 river profile to the mountain front, we may calculate the incision rate along patch 2 during stage 3, $I_{2,3}$ as

$$I_{2,3} = I_3 - Z_{b2}/t_3 \qquad (6b)$$

It is reasonable to assume that during the knickzone formation, the incision rate along patch 2 should be greater than present incision rate, $I_2 > I_{2,3}$. With known $I_3$, $t_3$, and $z_{b2}$ (see table S4 for details), the minimum rate of incision during knickpoint formation, $I_2$, is $50 \pm 17$ m/kyr (2 $\sigma$ confidence).

To find the maximum duration for the time of knickpoint formation, $t_2$, we introduce a second set of constraints, derived from the total incision at the mountain front since 9.5 kyr B.P. and the total incision along patch 1 since 6.9 kyr should both match the observed terrace record:

$$H_{9.5kyr} = t_1 \times I_1 + t_2 \times I_2 + t_3 \times I_3 = I_1(9.5\ kyr - t_2 - t_3) + I_2 t_2 + I_3 t_3 \qquad (7)$$

and

$$H_{6.9kyr} = I_1(6.9kyr - t_2 - t_3) + I_{1,2} t_2 + I_{1,3} t_3 \qquad (8)$$

In addition, the geometric relationship between patch 1 and 2 should also be satisfied. The effective base level of patch 1 ($z_{b1}$) is determined by the relative incision rate between patch 1 and patch 2, and patch 1 and patch 3, and the durations of the 2nd and 3rd stage. Based on equation 5, we have

$$z_{b1} = (I_2 - I_{1,2})t_2 + (I_3 - I_{1,3})t_3 \qquad (9)$$

Based on constraint 2 and $I_{2min}$=$50 \pm 17$ m/kyr, we calculate the maximum duration of the 2nd stage as $680 \pm 460$ yr (2 $\sigma$ confidence), which is approximately half of the duration suggested by the terrace record alone. It is worth noting that because the $t_3$ we used in the calculation is based on a minimum age for the lowest terrace, the maximum duration of stage 2 may be even shorter and the minimum stage 2 incision rate at mountain front may be even faster than calculated here. In addition, though we are able to constrain the maximum duration of the 2nd stage to ~700 yr, which suggests this <700 yr event happened sometime between 4.7 to 3.2 kyr B.P., we do not have enough data to pinpoint when this event occurred within this 1500 yr time span.

To verify these model results, we ran simulations of the channel evolution using our parameter estimations and compare this with our surveyed channel profile (Figure 8). The simulation results fit well with the observed channel profile, while the modelled channel positions at the end of stage 2 have a relatively wide range of possible geometries because we have less control on the starting and ending time of stage 2. The simulation results therefore verify that the 680 yr maximum duration and 50 m/kyr minimum 2nd stage incision rate are good constraints for the Beida River incision history.

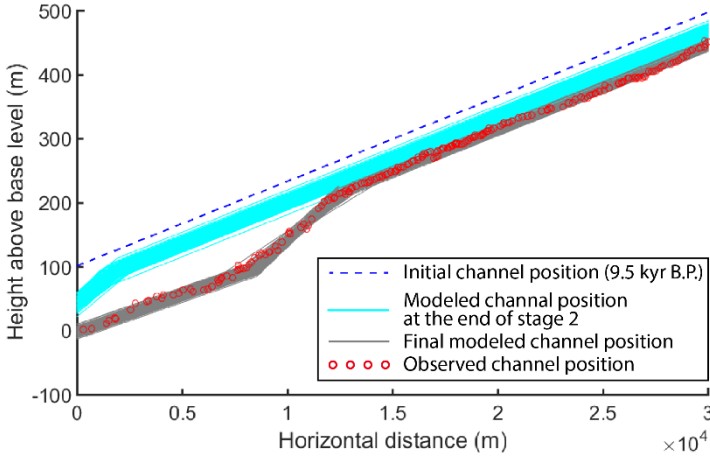

**Figure 8 Monte-Carlo simulation of channel position at the end of stage 2 and at present, end of stage 3, compared to the observed river profile. 500 simulations results shown. Parameters for simulations were selected from valid model results for the timing, duration, and incision rate stages 2 and 3.**

410

### 5.4 Climate implications of the Beida River incision history

Based on our prior study (Wang et al., 2020), the major T1 and T2 fill terraces of the Beida River correspond to the last glacial and penultimate glacial periods, respectively. A similar pattern occurs along the Hongshuiba River, the next major river to the east of the Beida River. The T1 terrace here is similar in age as T1 at the Beida River (26.5±5.0 kyr in Yang et al., 2020;

415    22.5±2.2 kyr in Hetzel et al., 2019), and the T4 terrace here has similar age as the T2 terrace at the Beida River (153±12 kyr in Hetzel et al., 2019). In the eastern North Qilian Shan, terrace records of the Shagou River also show a fill-cut pattern synchronized to glacial-interglacial cycles (Pan et al., 2003). Therefore, we suggest the formation of fill terraces along the major rivers draining the North Qilian Shan is controlled by the same mechanisms, linked to glacial-interglacial cycles. A full fill-cut cycle begins with the river backfilling its canyon with sediment during the glacial period. This is a period with elevated,

420    glacially derived sediment flux, and perhaps a drier climate and lower discharge than at present. At the glacial-interglacial transition, the river transitions from deposition to erosion as sediment supply gradually declines. This corresponds to the first incision phase of the Beida River (24-9.5 kyr B.P.). The fast, second incision phase of the Bieda River starting after 9.5 kyr B.P. corresponds in time with highstands of several endorheic lakes located in the Northwest China including the Juyanze paleolake, which have been interpreted as a signal of intensification of monsoon influence (i.e., Herzschuh et al., 2006;

425    Hartmann and Wünnemann, 2009; Jiang et al, 2008; Li et al, 2009; Rhodes et al, 1996; Wang et al, 2013). Therefore, the post-9.5 ka fast incision phase may relate to a regional increase in discharge, beginning near the end of the last glacial termination, as a result of the introduction of monsoon-derived summer moisture.

The mid-Holocene <680±460 yr event which punctuated the moderately fast incision of the Beida River since 9.5 kyr B.P. seems to be a result of an unusual short-term hydrologic perturbation. Similar knickzones found in other western North Qilian rivers, i.e., Hongshuiba River and Maying River (Figure 1), suggest that this was a regional event that affected the entire western North Qilian Shan. We suggest this short-term hydrologic anomaly was driven by an increase in precipitation as a result of increased monsoon influence, and possibly enhanced by glacial melt due to warmer temperatures. Records of the terminal Juyanze paleolake suggest the highest mid-Holocene lake level occurred at 4.2 kyr B.P. (Hartmann and Wünnemann, 2009), and pollen records of the same area suggest a wet, or pluvial period between 5.4 and 3.9 kyr B.P. (Herzschuh et al., 2004). Regionally, evidence for similar humid periods can also be found from Zhuyeze, a lake fed by Shiyang River of the eastern North Qilian Shan (Chen et al., 2006), Qinghai Lake (Chen et al., 2016) located within the southeast Qilian Shan, Tianchi Lake of Liupan Shan (Zhou et al., 2010), and Yanhaizi Lake of Inner Mongolia (Chen et al., 2003). Cave records from upper Hanjiang region and Qinling Mountains (Tan et al., 2018) and stratigraphic sections from the Loess Plateau also support the existence of a mid-Holocene humid period (Fang et al., 1999; Fang et al., 2003; Chen et al., 1997; Xiao et al., 2002). All of this evidence suggests there was a humid period during the mid-Holocene that correlates well with our stage 2 rapid incision period. Considering the western North Qilian Shan is located at the transition zone between the Southeast Asian Monsoon and mid-latitude westerlies, and that wet periods regionally correspond to increased monsoon influence (Chen et al., 2018; Tan et al., 2018), we hypothesize that the <700 yr incision event on the Beida River occurred at the peak of monsoon influence within the region.

In summary, combining all the circumstantial evidence documented here, we suggest the Beida River experienced the following incision and climate history since 24 kyr B.P. After 24 kyr, the river started to incise as it gradually depleted its sediment supply during the deglaciation and early interglacial period. Its incision rate (corrected for uplift) of 1.0±0.8 m/kyr was more or less in balance with the rock uplift rate during this time. During the Early Holocene, the humid Southeast Asian Monsoon expanded to the central North Qilian Shan, where it affected the Hei He drainage and filled Juyanze lake to a high lake level. The expanded monsoon correlates to the 1$^{st}$ incision stage of the Beida River when the canyon incised at a rate of 5.6±2.2 m/kyr. During the mid-Holocene, the monsoon influence grew stronger, leading to a regional pluvial period. This peak of monsoon influence lasted less than 680±460 yr in the Beida River drainage, which led to an increase of precipitation, and therefore an increase of water discharge and incision rate. This climate event led to deepening of canyons into the foreland basin, and prompted the formation of bedrock knickzones in the Maying, Hongshuiba, and Beida River sub-basins of the Hei He. During this pluvial period, the incision rate of the Bieda River increased to at least 50±17 m/kyr. Afterward, the discharge decreased to its present condition and river incision slowed to 11.6±1.4 m/kyr (Figure 9).

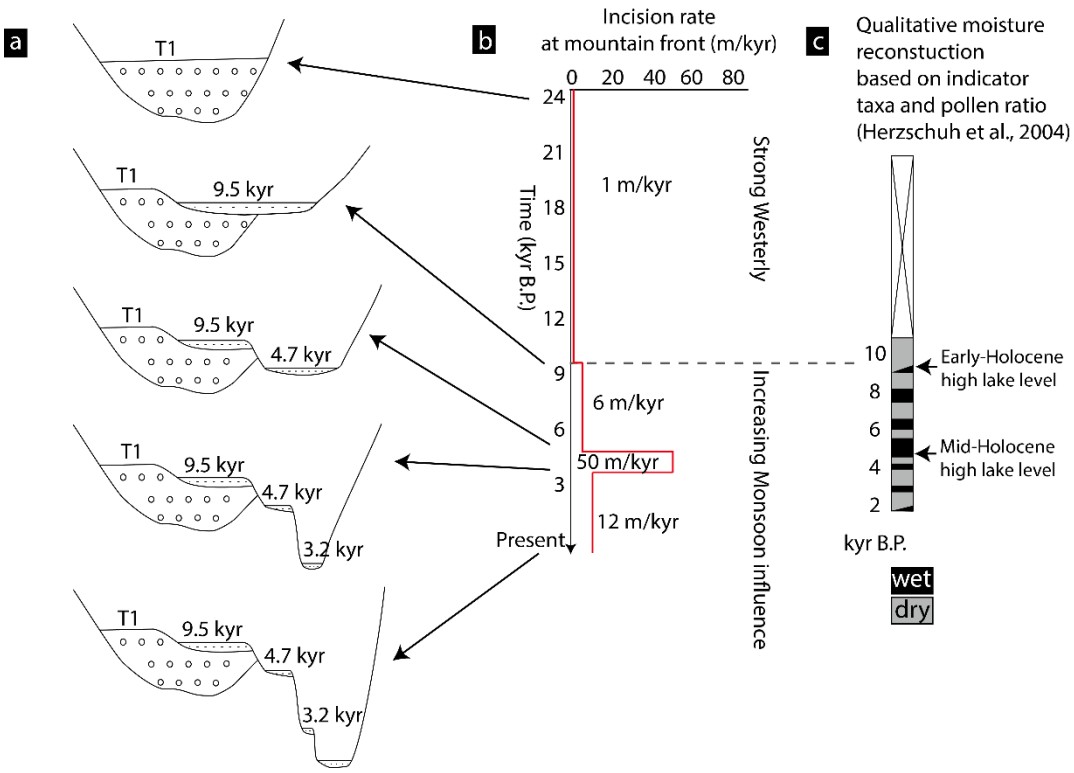

**Figure 9 a) Schematic cross sections of Beida River channel evolution at the mountain front; b) diagram of incision rate vs. time since 24 kyr B.P.; c) climate reconstruction based on the Eastern Juyanze palaeolake record (Herzschuh et al., 2004).**

Though we hypothesize that knickzones present on the Hongshuiba River and Maying River also formed at the mountain front during a mid-Holocene pluvial period, the consistent locations of these knickzones 7-13 km upstream of the mountain front (Figure S7a(1), b(1), c(1)) is somewhat surprising, considering the dissimilar sizes of these drainage basins (Table 1). One might expect that the knickzones on the Hongshuiba River and Maying River would have retreated less, as these drainage basins are smaller than that of the Beida River (Table 1). Normalizing by upstream drainage area to produce a χ plot (Perron

and Royden, 2013) of these rivers (Figure S7a(2), b(2), c(2)) confirms that the knickzone location on the Maying River has retreated about twice this normalized distance upstream relative to the Hongshuiba River. Likewise, the knickzone on the Hongshuiba River is about 1.5 times this normalized distance upstream relative to the Beida River. We suggest this pattern of greater knickzone retreat from west to east is partially the result of the strong west to east precipitation gradient present within western North Qilian Shan. Normalizing by the ratio of discharge per unit drainage area at the mountain front (Table 1), relative

to the Beida River, partially accounts for the difference in knickzone positions on the Hongshuiba River and Maying River (Figure S7a(3), b(3), c(3)). It is worth noting that, though precipitation gradient related to elevation has the potential of further reducing the normalized knickzone retreating distance of the Hongshuiba and Maying River, we are unable to incorporate this effect into the adjusted χ plots due to lack of detailed precipitation data. Both the timing and amount of precipitation during the mid-Holocene pluvial period could account for the remaining difference. The mid-Holocene highstand of the terminal

Juyanze paleolake precedes the very rapid incision of the Beida River by a few hundred years (Figure 9, Herzschuh et al., 2004), which suggests that the pluvial period may have begun earlier in the eastern part of the Hei He drainage basin. This is consistent with the observation that both the Hongshuiba and Maying River canyons are incised deeper into the foreland basin than the Beida River, indicating an even faster integrated rate of Holocene incision. It is also possible that the west to east precipitation gradient was stronger during the early and mid-Holocene than at present. Because knickzone retreat rate will be

faster for steeper, narrower channels (Finnegan et al., 2005), the rate of abrupt base level drop should affect the rate of retreat of these knickzones. Contrary to the large rivers, lesser streams draining across the mountain front and the Beida River tributaries do not exhibit evidence for deep, rapid incision. We attribute this difference to the aridity at lower elevation and that only those drainages that tap high elevations (over 4000 m) are supplied continuously during the summer by glacier melt. In summary, variations in discharge, due both to monsoon influence and orographic effects, strongly impacts the Holocene

incision rate and evolutionary pattern of both major and small rivers draining the western North Qilian Shan.

**6 Conclusions**

The Beida River in the North Qilian Shan has incised deeply into both the bedrock and the adjacent foreland basin sediments. The incision rates indicated from terrace records and our models greatly exceed rates of tectonic uplift here. Our work demonstrates the capability of bedrock rivers in arid regions to incise deep channels and form fast retreating knickpoints within

a short period. Field investigation and geomorphic mapping identify a 24 kyr fill terrace, T1, and several sets of inset terraces below. The longitudinal profile of the present river channel preserves a steep knickzone, located 10 to 12 km upstream of the mountain front. Terrace ages, and relationships between terrace treads and the riverbed, indicate that the knickzone was formed quickly after 4.7 kyr BP, likely driven by an increase of river discharge. By applying the concept of slope patches along with channel geometry and terrace records, we constrain that during the knickzone formation, the incision rate was at least $50\pm17$

m/kyr and the duration of this period of increased discharge was less than $680\pm460$ yr, which is about half of that estimated from the sparse terrace age record alone. The period of increased incision rate identified from the Beida River correlates to a pluvial period recorded at the terminal Juyanze lake. The likely cause of rapid incision of the Beida River, and adjacent rivers with similar deeply incised canyons, is the increased influence of the Southeast Asian Monsoon over the Holocene, with the most rapid incision period corresponding to a peak of monsoon influence ca. 4 to 5 kyr B.P.


**Acknowledgments, Samples, and Data**

This work was supported by the US National Science Foundation [grant number EAR-1524734] to Michael Oskin, the National Natural Science Foundation of China [grant number 41571001] to Youli Li, the Second Tibetan Plateau Scientific Expedition and Research (STEP) [grant number 2019QZKK0704] and the National Natural Science Foundation of China [grant number

41622204; 41761144071] to Huiping Zhang, and through Cordell Durrell Geology Field Fund to Yiran Wang. We placed our data in a permanent repository at Open Science Framework (https://osf.io/bpvw9/?view_only=da77dd9c7ace4e76bf96dedc9fbaa61f), and also upload as a zip file separately.

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
