# Peer review of "Rapid Holocene bedrock canyon incision of Beida River, North Qilian Shan, China"

_Earth Surface Dynamics, 2021_

## Referee Comment (RC3)

Review of Wang et al., Rapid Holocene bedrock canyon incision of the Beida River, North Qilian Shan, China for consideration in Earth Surface Dynamics

**Summary of study:**
Wang et al. use river terraces, geochronology (previously generated cosmogenic depth profiles and new [14]C and luminescence data), and river profile analysis to understand the late Quaternary incision history of the Beida River. The authors map several large fill terraces (previously dated with cosmo) and three inset terraces, described as strath terraces, that are dated using [14]C measurements of charcoal in and luminescence geochronology. The lowest of the large fill terraces (~24 ka) prograndes and merges with an alluvial fan beyond the active mountain front. The three inset terraces are mostly, but not exclusively, found upstream of the active range front and are inferred to have been abandoned at ~9.5, 4.7, and 3.2 ka. Upstream of the mountain front, the river profile shows a large (bedrock?) knickzone. Field observations suggest that after deposition of the 24 kyr fill terrace, the river cut into bedrock, rather than reoccupied the same channel position, a history recorded by the inset terraces. The age and elevation of the inset terraces indicates rapid Holocene incision into bedrock that far outpaces tectonic uplift rates determined from the older, deformed fill terraces by Wang et al. (2020). The authors hypothesize that this rapid pulse of incision is related to the generation of the now headward migrating knickzone. Based on the incision data and using a slope-patch approximation (*sensu* Royden and Perron, 2013), the authors predict paleo-river bottoms and incision histories, absent tectonic uplift. Based on all of this evidence and analysis, the authors argue that the rapid Holocene incision is the result of climatically-induced changes in water discharge and sediment flux that caused the river to rapidly downcut. The formation of the knickzone is argued to be the result of this downcutting as the contrast in erodibility between the erodible alluvial fan downstream of the mountain front and the relatively more resistant bedrock upstream of the mountain front.

**Major comments and recommendation:**
I think that this is an interesting study that is well suited for publication is E-surf. The study presents new mapping and geochronology and puts them together in a nice way that combines field results with a simple model for bedrock river incision. The results are interesting and highlight the role of climate and variable substrate resistance to erosion as key players in determining incision rate, bedrock channel morphology in response to external perturbations. That said, I think that the presentation can be cleaned up, which will benefit the impact of the study. Specifically, I think the quality of the figures needs to be improved, and perhaps a few figures can be added to help guide the reader through the observations more easily. I also think a little restructuring might help improve the flow of the manuscript. There are some aspects of the analyses that I found unclear and some additional analyses that could be performed to help better support the interpretations forward in the study. Below I highlight these and some other key points that I think need to be addressed before publication.

*(1) Figures:* Many of the figures are too small and difficult to follow. Some figure additions are needed (e.g., terrace stratigraphy in key locations as in Wang et al. (2020) Figure 4b,c and additional river profiles in the region). I also have several specific suggestions that I think can help improve the presentation of the figures, as well as a couple of suggestions for additional figures. I include the specifics in my detailed line-by-line comments below.

*(2) Structure:* I struggled quite a bit with the structure and flow of the manuscript. I think that part of this has to do with the legacy of revisions of a previous version of the study. The motivating observation is the observed knickzones in the river profiles, but the key data is the incision history from the terraces. A cleaner presentation of the results might focus on the terraces first and then discuss the

characteristics of the river profile in the context of the incision rates and patterns. This could also serve to streamline the discussion because the terraces record the incision pulse, which presumably generated the knickzone due to variable substrate erodibility. Currently, from the results onward, the terrace and river profile discussions are mixed together, making a really interesting story challenging to follow. I think that this mixing of analyses and discussion makes it more difficult for the reader to clearly visualize the key points of the conceptual model for knickzone formation – climate-driven incision pulse causes downcutting in the alluvial fan downstream of the mountain front and an upstream change in substrate erodibility results in the formation of the knickzone. This is in the paper, but it could be presented more clearly and concisely.

*(3) Support for the preferred interpretation (and maybe some additional restructuring):* I think the authors can rely more on the geochronology to support their favored interpretation. For example, the emplacement T1 fill terrace corresponds (roughly) with rapid cooling around the LGM and T2 the timing of the preceding full interglacial. This information is not new but presented in Wang et al. (2020), so it is unclear to me why the apparent correlation between fill terrace deposition and climate isn't presented in the background section. Doing so would place the link between climate and incision-aggradation cycles in the reader's mind early and key them into thinking about temporal links between the terrace stratigraphy and climate as they read through the rest of the manuscript. Additionally, I think that the authors can more strongly interpret the timing of inset terrace deposition with regional and global climate records and that this would help their arguments. For example, T1' is abandoned roughly at the Preboreal-Boreal transition. This seems like a missed opportunity in the discussion.

Furthermore, in looking at Google Earth, I think observations from adjacent drainages can be used to the benefit of the climate and discharge interpretations. Importantly, it is evident from Google Earth that the only rivers that incise into the alluvial fan/fill north of the range front are those that drain high elevations. This is clearly recognized by the author's and mentioned in their interpretations, but what is missing are observations from satellite imagery (or Google Earth) and river profiles (not just from the trunk channels) from adjacent rivers to help support this kind of interpretation. This would be a small added figure/analysis but could help convince a skeptical reader.

*(4) Slope Patch:* I was a little confused by some aspects of the slope patch approach used here. I understand the formulation in Royden and Perron (2013), and I see what the authors are trying to do. My main question has to do with the observation of variable channel width that is not included in the analysis here. In the slope patch formulation in Royden and Perron (2013), they use the general stream power model ($E = KA^mS^n$). This equation presumably accounts for/assumes hydrologic (Q) and hydraulic (W) scalings with A encapsulated by exponent m. To handle these along stream changes, Royden and Perron (2013) use a coordinate transformation of distance, introducing the non-dimensional distance chi. From chi, along with other non-dimensionalization, Royden and Perron (2013) come up with their elegant slope patch solutions.

As I understand it for this application, Q is assumed to be uniform (or approximately so) over the length of the study area; however, W is shown to vary. It seems that equation 3 only works as applied in the manuscript if W is also uniform with only S and I varying along the channel length. It seems like the observed variations in W make it difficult to apply this simplified version of the slope patch approach as presented in equations 2-5.

That said, I do think that this analysis is useful, but I think that the caveats and assumptions made in this simple analysis need to be discussed and explained fully. As written, I think many readers might not realize that this analysis is limited to channel sections where Q and W are assumed to be uniform or that one needs independently constrained incision records for this kind of analysis. These requirements make the approach limiting and not general. I think these limitations/data requirements needed to be more clearly articulated. I also think the slope patch discussion could be rounded out by linking the modeled changes in incision rate and river profile geometry through time back to the observed slope patches in section 5.2. For example, how close are the observed and modeled slope patch gradients when run to the modern?

I realize this is a long comment that can probably be handled with the addition of a few sentences in the methods and discussion; I just wanted to make sure that I completely communicated my questions and tried to articulate where and why I was a little confused.

*(5) Terrace stratigraphic relationships:* It is really hard to understand the general terrace stratigraphy without photos and figures. One suggestion is to add a composite terrace stratigraphy for several locations along the river to illustrate the relationships between the fill terraces and the inset terraces. What I have in mind is something like figure 4b and c in Wang et al. (2020). It was hard for me to understand the context of the terrace observations without a figure like this. Also, from figure 4 in Wang et al. (2020), it looks like the T1' terrace is a cut terrace (meaning it is cut into the T1 fill terrace) and is not a strath terrace. Is this correct? If so, does it become a strath terrace up valley? Also, how might this play into the interpretations forwarded in the study (i.e., strath vs. cut terraces)?

In summary, I think this is a cool study, well suited for publication in E-surf. With some moderate revisions, the study can be strengthened, and the arguments in support of the preferred interpretation strengthened.

If any of my major or detailed comments are unclear, please reach out for clarification.

Best,

- Sean F. Gallen

Detailed line-by-line comments.

L 13: incision accelerated from what value to 25 m/ky?

L 14: "faster" rather than "larger"?

L 17: "We interpret that this period of increased…"?

L 28-29: The role of lithology is critical here and probably deserved more discussion in section 5 somewhere. Without the change in erodibility due to the lithology change at the range front, the knickpoints would be causing the profile to relax (assuming elevated incision is due to an increase in Q, as suggested). It seems worth explaining this to the reader in some detail later.

L 43 [Figure 1]: Make the map of figure 1 larger, label some of the key geographic features mentioned in the text on the map (e.g., Hexi corridor), provide a zoomed-in shaded relief map of the location with the three highlighted rivers, and combine the current figure 1 and 2. Doing this would introduce the reader to all of the general key observations without having to flip back and forth and look in google earth to understand much of what is written. This might also be a good location to show some google earth images of the studied and adjacent rivers (see part of my major comment 3 above). Also, are the glaciers in this map active or from the Pleistocene?

L 51 [Figure 3]: The photos in figure 3 are very small but really important. Please make them bigger. It would also be useful to add some photos showing the key relationship that the river didn't reoccupy the same valley filled with T1.

L 66-72: Is there any information on Pleistocene ELAs or glacial extents in this region? It might be relevant.

L 85-94: The typical convention in name/labeling terraces is that the older units have lower numbers and the younger units have higher numbers. This is the same convention as bedrock map units. It appears that in this study and in Wang et al. (2020) the opposite is used (older have higher numbers than younger). Because of this, I found this section very confusing the first time I read it until I saw figure 5. Also, why not label the inset terraces with something like letters? The lack of naming for them makes reading a little awkward.

L 99: I can't see the reverse fault offsets in figure 1.

L 114: inset rather than insect.

L 119 [Table 1]: Can a row be added to link these ages to the terraces they were collected from? Having a composite terrace stratigraphy figure preceding this table and naming/labeling the inset terraces will make this pretty easy to do.

L 124: see my major comment 4 regarding this section.

L 128: Need to link A and Q in some kind of statement.

L 136: km^2

L 137: km^2

L 162-164: These changes in width impact hydraulic geometry and thus shear stress imparted by a flow of a given magnitude. How might these changes in channel width impact the assumptions made in the simplified slope patch calculations?

L 171-182: This section would be easier to follow if the inset terraces were labeled or named, and there were some figures showing their composite stratigraphy in a few key locations along the river.

L 184 [Figure 4]: can the y-axis label be oriented as in figure 2 with the labels reading from bottom to top? Also, I don't understand panel b, and I assume the y-axis is mislabeled (should be elevation rather than width). How does the channel bed have negative elevation?

L195-198: Photos of these critical relationships would really help!

L 204 [Figure 5]: I don't find panel **a** very helpful because the fill terraces (besides T1) aren't the main part of this manuscript, but the inset terraces are. I can't see the distribution of the inset terraces here with respect to T1, so I don't get much out of this panel. Panel c should be enlarged to make it easier to read.

L216-218: What about tectonic subsidence? No correction is needed for that?

L235-237: It would be nice to see some images of this from Google Earth or something. These observations are very helpful to the interpretations presented here.

L239: Be careful here. The change in erodibility of the alluvial fan and the bedrock is essential to explain the knickpoint. The driver of incision might be a temporal increase in Q, but if that occurs in uniform substrate, the river profile will relax, forming an "inverted" knickpoint that migrates upstream. Here, the inferred elevated Q forces the river gradient to relax in the alluvial fan (incision), and this base level fall steepens the "harder" bedrock rock reach upstream. The interaction of the inferred climate change along with the spatial change in substrate erodibility explains the observations. Both are required.

L245-249: Yes, changes in substrate erodibility are needed as stated here.

L305-324: It seems like this would be better placed before the slope patch discussion. It also makes me wonder about the utility/generality of the slope patch approach used here. The Royden and Perron (2013) formulation implicitly takes changes in width into account via the calculation of chi, here that doesn't work. It might be helpful to highlight this point earlier in section 3.3 and mention that this point will be discussed in detail in section 5.

L 325: It seems like showing the relaxation of the river profiles beneath the incised alluvial fan surface for a few of the rivers would help support the point that there is a reduction in the river channel gradient beyond the range front. In the framework used here, the two most obvious explanations are changes in water discharge and/or sediment flux. Considering that the rivers that show this behavior all drain high elevations, the interpretations forwarded here seem reasonable to me. The key observation that isn't explained well in my mind is that the river gradients have declined downstream of the knickzones in the alluvial fans. The gradient decline, coupled with the change in substrate erodibility, generates the knickzones, correct?

L 378-395: I am left wondering if it might be helpful to explain the possible links between climate and the previously dated T1 and T2 fill terraces earlier in the study. This is all based on published data and will help establish a link between climate and geomorphology in the study area before diving into the inset terraces. It can be revisited here, but much of this section is related to previous studies and knowledge gained, so it seems better suited for the background.

---

## Referee Comment (RC4)

Dear Editor and Corresponding Authors,

It was a pleasure reading this contribution from Wang et al. The manuscript uses field mapping, [14]C, and OSL data collected from river terraces to constrain the incisional history of the Beida River since ~24 ka. The resulting chronology is then coupled with a numerical incision model to constrain the duration and magnitude of an incision rate increase responsible for creating a prominent knickzone on the river. Finally, the authors argue that variations in the Beida River incision rate that have occurred during the Holocene were primarily driven by increases in precipitation related to the increasing influence of the Southeast Asian Monsoon. The incision rate began to increase at ~9.5 ka due to the expansion of the monsoon into the North Qilian Shan, and a higher intensity increase responsible for creating the knickzone occurred sometime after ~4.7 ka due to strengthening of the monsoon. The numerical model suggests that the knickzone was created over a maximum duration of ~700 years at a minimum incision rate of 50 m/yr.

This is a very interesting study containing novel geochronology data, modeling techniques, and insights into the relationships between climatic forcing, lithologic variations, and river incision. I believe that the manuscript can be suitable for publication after some minor revisions. I have only one suggestion that rises to the level of a major comment concerning the veracity of the implied model result, and I think that all my comments can be addressed without any major modifications to the overall structure of the manuscript. I'd also like to echo the suggestion from RC1 and RC2 that the authors include chi plots of the Beida River and its tributaries to strengthen their argument.

Overall, I find this study to be an extremely interesting example of how regional climate forcing can significantly impact the pace of river incision, and I look forward to reading the revised manuscript. If the authors have any specific questions regarding my comments, please feel free to contact me.

Chris Sheehan
Boston College Department of Earth and Environmental Sciences
sheehacz@bc.edu

Major Comment:

My only major comment concerns some of the assumptions that go into the coupled incision model and the discussion of its results. The model assumes a simplified Beida River incisional history occurring in three stages: Stage 1 (~9.5 – 4.7 ka, ~5.63 m/kyr), Stage 2 (~4.7 – 3.2 ka, ~18.6 m/kyr), and Stage 3 (~3.2 ka to present, ~11.56 m/kyr). Using the incision model, these discrete stages suggest that the knickzone was created over a duration of ~700 years with a minimum incision ate of ~50 m/kyr. The actual incision history was likely much more complex, but I think that simplifying it into three stages for the sake of this analysis is perfectly justifiable.

However, I am curious to know how slightly different yet plausible incision scenarios might affect the model results. In particular, the boundary between Stages 2 and 3 is inferred to be 3.2 ka. However, this is only based on a single datapoint (a single OSL sample from sampling site B). One could imagine a more complex incisional history between 4.7 ka (the boundary between Stages 1 and 2) and present. For example, perhaps the incision rate has gradually decreased since ~4.7 ka, implying that Stages 2 and 3 would display an exponential curve on Figure 7 rather than two linear segments. Alternatively, perhaps the terrace at site B was abandoned much later than implied by the ~3.2 ka depositional age, justifying the inclusion of a fourth Stage in the model.

I think the authors should add a few sentences in Section 5 that discuss how their model results might vary under different circumstances. Importantly, they should address how their conclusion that the knickzone was created over ~700 years at a minimum rate of ~50 m/kyr might vary. I don't think they need to perform a full sensitivity analysis (though that would be interesting!), nor do they need to quantify the duration and magnitude of knickzone formation under specific, alternative conditions. Rather, I think they should just briefly list some plausible scenarios that could either increase or decrease the implied duration of knickzone formation and qualitatively discuss the effects. I think this could be done just by adding a few sentences to Section 5.1.

I listed this as a major comment because the 700-year duration and 50 m/kyr incision rate are a major takeaway of this study (I expect that they will immediately grab the readers' attention in the abstract). Therefore, I think it is very important to put them in context.
* * *
Minor Comments:

1.  I think that Table 1 could benefit from two changes. First, it is extremely difficulty to tell which sample name the 1-sigma and 2-sigma results correspond to. I recommend realigning the data in the calibrated age column so that their corresponding sample names are unambiguous. Second, I recommend adding a column showing the elevational position of each sampling site relative to T1. This will allow the readers to quickly reference the relative age of each terrace.

2.  I have an issue with one of the authors' arguments against the knickzone being a lithologic feature. I do agree with the authors' interpretation that the knickzones on the Beida River and its neighboring rivers were created by climatic forcing and are not likely the result of lithologic variations (i.e. variations in K in equations 1-3). The strongest evidence ruling out a lithologic origin is the lack of a knickzone preserved on the T1 strath (lines 240-241) and the continuous projection of terraces across patch 2 (Figure 6).

    However, consider this excerpt from Lines 236-239: **"In the neighboring Maying and Hongshuiba River, similar to the Beida River, present river channels also incised into Late Pleistocene fill terraces..., forming prominent knickzones 10-15 km upstream of the mountain front (Figure 2). This suggests these knickzones share similar origins, reflecting regional forcing. Local variations of lithology would thus be an unlikely cause for knickzone formation."** If I am interpreting this correctly, the authors argue that if the Beida River knickzone was a lithologic feature,

we would not expect to see similar knickzones on the Maying or Hongshuiba rivers because the lithologic variations along the Beida River would not likely be present along the other two rivers.

I disagree with this reasoning. I am unfamiliar with the regional geology of the North Qilian Shan, but I can see on Google Earth that there are SW-dipping lithologic contacts along the Beida River corridor. These contacts are roughly parallel along their NW-SE strike, and so without a more detailed geologic map, it seems entirely reasonable that the Beida, Maying, and Hongshuiba rivers might cross the same bedrock units in each of their knickzones. Also, it appears on Google Earth that the Beida River crosses lithologic contacts (marked by stark color contrasts) at the transition from patch 1-2 and patch 2-3.

I recommend that the authors examine the spatial relationships between the Beida River patches and the underlying lithologic contacts. I still agree with their interpretation that the knickzone is most likely a transient feature created by climatic forcing, but if it happens that the knickzone is underlain by a low-erodibility rock type, then this might be a contributing factor worthy of a brief discussion. Alternatively, if the boundaries between the three patches do not correspond with major lithologic transitions, then this will strengthen their interpretation. The authors could also extend this analysis to the other two rivers.

Since I am unfamiliar with this region, I do not know if the authors can obtain a geologic map with enough detail to perform this analysis. If they cannot, I would be satisfied if they just removed their argument in Lines 236-239 and relied on their evidence in Lines 240-241.

3.      Between Lines 363 to 371, it is unclear to me whether some sentences are information from Tan et al., 2018 (cited in Line 362) or the authors' own interpretations. I think that these require clarification. Specifically, these lines:

- Lines 363- 364: **"Between 24 kyr and 9.5 kyr B.P., the Beida River drainage was under the dominant influence of the arid westerly moisture source."**
- Lines 365-367: **"During the Early Holocene, the humid Asian monsoon expanded to the central North Qilian Shan, where it affected the Hei He main stem and filled Juyanze lake to its highest lake level."**
- Lines 368-371: **"During the mid-Holocene, the Asian monsoon grew stronger, starting in the Hei He drainage around 5.4 to 5.1 kyr B.P., and then expanding further to the western North Qilian Shan a few hundred years later. This peak of monsoon influence lasted less than 700±340 yr, which led to an increase of precipitation therefore an increase of water discharge and incision rate."**

If previous research demonstrated that the Asian monsoon expanded into the North Qillian Shan in the early Holocene and then strengthened during the mid Holocene, then this independently supports the authors' conclusions that the knickzone was created by mid Holocene climatic forcing. However, it is unclear to me whether this was previous research or a novel idea presented here.

4.      In Lines 335-346, the authors argue that a discharge contribution from melting glaciers would not have been enough to trigger knickzone formation. While this intuitively seems correct to me, I'm not sure that I agree with their reasoning. They cite metrics of modern glacial melt and discharge contributions, compare these to possible conditions in the past, and conclude that mid Holocene melt contributions could not have triggered knickzone formation. However, the authors have not quantified the actual discharge increase necessary to create the knickzone. Without this information, I don't think that the other metrics presented here definitively rule out glacial melt as the primary driving mechanism. I recommend modifying the sentence in Lines 345-346 to clarify this.

5.      Can the authors comment on how the long-term (i.e. averaged over several glacial / interglacial cycles) bedrock incision rate may compare to the tectonic uplift rate? Beida River incision clearly outpaces uplift during the transition from glacial to interglacial conditions, but averaged over $10^5$-year timescales, is the river more or less in steady-state? Quantifying this long-term trend requires dating

several generations of strath terraces (see Pederson et al., 2006), and so this may not be possible to do for the Beida River. Still, it might be useful for the authors to add a few sentences discussing this in Section 5.
* * *
Line-by-line comments:

- Line 114: "insect" should be "inset".
- Line 128: "A is drainage area" is unnecessary. Drainage area does not appear in this particular formulation of the stream power equation (Q is used instead), and so the variable "A" does not need to be defined.
- Line 135: "drainage area of Beida River" should be "drainage area of the Beida River".
- Lines 136 and 137: "km2" should be "km$^2$".
- Lines 158, 245, and 384: "Beida River" should be "the Beida River".
- Lines 169 and 170: "above present riverbed" should be "above the present riverbed".
- Lines 172-173: Is site C on the T1' terrace? If so, shouldn't the samples imply that T1' was abandoned AFTER 9.5 ± 0.16 ka (not prior)? The changes that I recommended for Table 1 could help clarify this.
- Lines 172-180: While reading this section, I found it difficult to construct a mental map of the terrace position / age relationship due to their heights being listed relative to different surfaces. This is later clarified very well in Figures 6 and 7, but those figures are neither shown nor referenced for another 5 pages. I think the changes I recommended for Table 1 will be very useful here, because the readers will have already seen these data, and they can flip back to the table for a visual aid if necessary.
- Lines 198 199: Does this mean that none of the inset terraces below T1' have exposed straths?
- Line 219: "Incision rate" should be "The incision rate".
- Line 220: I think there may be unwanted spaces after some periods in this line. I think "between 4.  7 to 3.  2" should be "between 4.7 to 3.2".
- Line 229-230: Tying into my previous comment concerning long-term incision rates, over what period of time is the average uplift rate ~0.6?
- Line 246: "It is likely that knickzone" should be "It is likely that the knickzone".
- Line 381: "correspond" should be "corresponding".
- Line 382: "dating of T2 terrace" should be "dating of the T2 terrace".
- Lines 382-384: This sentence is awkward. I recommend breaking it into two sentences and changing "similar to that T1" to "similar to how T1".
- Line 387: "similar age as T2 terrace" should be similar age as the T2 terrace".
- Line 388: "Shagou River" should be "the Shagou River".

---

## Author Comment (AC1)

**Replies to comments**

Thank you for reading through our manuscript and sharing your thoughts! We are encouraged by your positive feedback on our research, and we are certain that your suggestions will guide us to refine our manuscript. We are confident that we can address your concerns with the revision, and we believe that after the revision the quality of the manuscript will be greatly improved.

Following are the detailed responses.

Sincerely,

Yiran Wang

**Reply to Dr. Wolfgang Schwanghart:**

*1. The three rivers shown in Fig. 2 have in common that they have a prominent knickzone as well as that they incise into the foreland-basin. The along-river distance of the knickzone to the mountain front is in all cases similar. According to the stream power incision model, however, slope patches should move upstream at a velocity dictated by upstream area. Hence, the three rivers should have similar upstream areas. However, looking at the map, the three sites seem to have very different areas. A way to address the effect of variabe drainage areas is to calculate chi as a horizontal along-river distance. This might be very helpful because it would provide additional evidence for (or against) a common base level drop, if all three knickzones should have similar chi values measured from the mountain front.*

Thank you for pointing out the interesting relationship between the knickzones among the major rivers. We have tried using chi plots to analyze these rivers, but eventually chose to use elevation versus distance plots for the following reasons. First, discharge is not solely correlated to the distribution of drainage areas within these catchments. Specifically, the annual precipitation is higher (>300 mm/yr) at high elevation, but very low (~100 mm/yr) at mountain front and foreland (Chen et al., 2018). In addition, tributaries at higher elevation (over 4000 m) are supplied continuously during the summer by glacier melt, while tributaries close to the mountain front are ephemeral streams with no glacial coverage. Second, there is a strong east-west precipitation gradient for the North Qilian mountain range. For example, the annual precipitation at the mountain front decreases from 200-300 mm/yr in Maying River drainage to 100-200 mm/yr in Beida River drainage (Geng et al., 2017). The above evidence suggests that in the western North Qilian Shan, topographic and geographic factors strongly influence the discharge of each of the major drainages, which not only makes drainage area a less effective proxy of water discharge, but also makes it difficult to compare different drainages by normalizing drainage areas.

In general, both slope and discharge contribute to knickzone retreat rate. Among the three major rivers draining the North Qilian Shan, each with incised foreland valleys and prominent knickpoints, the amount of incision varies by a factor of two. Specifically, at the mountain front, the height of the T1 terrace above the Maying River is ~250 m, whereas it is ~190m for the Hongshuiba River T1 and ~130 m for the Beida
River. These heights are controlled by the amount and rate of base level drop which is mainly controlled
by the incision process within the foreland. The foreland incision rate depends on both changes in
discharge and in sediment flux. For the Beida River, we have measurements of the incision rate from
stream terraces available to us, but the other rivers lack such information. Possible reasons why the
knickzone has retreated a similar distance on all three rivers thus include (1) differences in initial
knickpoint slope and the slopes of upstream/downstream patches, (2) higher precipitation in the east
versus west, and (3) possibly an earlier onset time for the period of most rapid incision in the east versus
the west. We will include this into a new section of the "Discussion", as we described below (L61-82).

*2. It is interesting to note that some of the rivers draining the North Qilian Shan have steeply incised in
the foreland while others have not, at least those that seem to have smaller drainage areas (I have not
investigated this in detail, though, so this may be a bit speculative). Looking at GEarth, it seems that in
particular those rivers with small catchments upstream the hanging wall are rather accumulating and
form large fans with no (at least to me) obvious trend towards incision. Where these rivers drain into the
gorges carved by the Beida river (and the other larger rivers), these smaller rivers are fluvial hanging
valleys and show signs of headward erosion. How is it possible that under a general increase in
precipitation, there are such different patterns in river incision? Or are there other processes responsible
for the incision? For example, could a lack of sediment connectivity and increased sediment storage
behind terminal moraines in the higher areas drained by the major rivers be a possible explanation for
sediment starvation of the main rivers and thus incision? At least, this explanation would not be at odds
with the data as well as your interpretation of the other terraces, as well.*

Regions at lower elevation and close to the mountain front receive the least precipitation, and have
little or no glacier coverage to provide steady summer runoff. Similar to the downstream tributaries of
the major rivers, small rivers draining the mountain front are ephemeral and dry for most of the time,
even during the summer monsoon season. Therefore, we suggest the overall low water discharge to
these low-elevation streams is the cause for the observed difference in incision pattern.

To address above concerns regarding incision of the major rivers (1) versus small streams and tributaries
(2), we plan to make following changes to improve our manuscript:

1. Include more climate background of our study area. Especially the precipitation distribution and
   how it contributes to water discharge of the rivers draining the North Qilian Shan.
2. Include more details on the major rivers (drainage area, annual discharge, canyon depth, etc.) in the
   background.
3. Add a new sub-section in "Discussion". In this section, we will compare knickzone retreat along the
   major rivers, and address the contrast with incision of tributaries and other small streams draining
   the mountain front. We will argue that these inconsistencies are related to the distribution of
   rainfall and glacially modulated runoff characteristic of the North Qilian Shan. Specifically, we will
   show that, first, drainage area is not a reliable proxy for water discharge in this region. Second, we
   will discuss that the Holocene incision of the bedrock rivers is controlled by incision of alluvial
   deposits in the foreland, which, in turn, is controlled by not only the water discharge, but also the
   sediment flux. These competing factors make it challenging to compare rates of bedrock incision
   across different drainages without independent chronological constraints. Such a comparison would make a useful test of the hypotheses developed from our Beida River study, but is beyond the scope of this manuscript.

*3. As noted in the previous comment, I think that the paper could benefit from some more geomorphometric analysis. So far, the authors mainly looked at the trunk streams of the rivers, and focused this analysis on the Beida River. In order to generalize their findings, however, some additional work is required that shows that the findings are consistent with the other drainage basins having these knickzones. One interesting observation that could shed additional light on these river systems is the analysis of the tributaries to these rivers. Looking at the DEM in Google Earth reveals numerous fluvial*

*hanging valleys tributary to the river downstream of the knickzone while they are missing upstream of the knickzone. Finally, comparison with incision rates measured elsewhere would help to judge the plausibility of the incision rates which are extremely high.*

Most of tributaries below the main stem knickzone form waterfalls where they join the main stem. This suggests when these tributary knickzones formed, the main stem was incising very fast so that the incision rate of the tributaries could not keep up with the main stem. This observation is consistent with terrace records and our model calculations. After the main stem knickzone swept through the downstream reach, the tributary knickzones may have not retreated upstream for several reasons. First, we argue that these ephemeral tributaries do not have enough stream power to incise fast into the bedrock over such a short time period (~4 kyr). Second, the extreme steepness of the hanging valleys and waterfalls prohibits the headward migration (i.e., Crosby et al. 2007).

Contrary to the referee's comment, tributaries of the Beida River above the main-stem knickzone do form smaller knickzones. However, these tributaries exhibit gentler slope where they join the main stem than tributaries downstream of the main-stem knickzone. We interpret that these upstream tributary knickzones reflect an ongoing process of adjustment related to the increase in incision rate of the main stem since 9.5 kyr. As shown by fig. 7 of the manuscript, incision rate along patch 1 increased at the beginning of stage 1 (9.5 kyr), and this rate may not have varied much from stages 2 to 3, as the steep main-stem knickzone that formed during stage 2 has not yet propagated past these upstream tributary junctions. The incision history we derive through a combination of terrace dating and modeling explains well this difference in tributary behavior upstream and downstream of the main-stem knickzone.

In our revised manuscript, we plan to include example tributary profiles (6 tributaries downstream of the main-stem knickzone, 6 tributaries upstream of the main-stem knickzone) in the "Results" section, and include a discussion of the tributary profiles in the "Discussion" section. We believe the tributary incision history provides additional evidence for our hypothesized mid-Holocene pulse of rapid incision.

*4. A short comment on the supplements: The quality of the pictures in the supplements is relatively bad. Consider storing the pdf with images in higher resolution.*

*5. A short comment on the figures: I partly found it difficult to read these figures. Perhaps I am getting old, but some of these figures (in particular photos) are very small.*

We apologize for the quality of these figures. We will make sure to replace these with higher quality
versions in our revised manuscript.

**Reply to Dr. Richard Ott**

*My pain point is that a more thorough morphometric analysis would benefit the manuscript. The authors*
*argue for a precipitation increase which should be regional. Therefore, it would be good to have chi-plots*
*of the Beida and neighboring rivers, including tributaries, for the lower river section where the*
*knickzones are located. Are the knickzones migrating up the tributaries, too? Are they at the same chi-*
*distance compared to the trunk rivers? If, e.g the knickpoints do not manage migrate up the tributaries*
*as fast the trunk (in chi-space), this could be an indication that glacial melt is indeed the controlling*
*factor, whereas an even increase in precipitation should affect all streams in the region in the same way.*

Thank you for pointing out the interesting relationship between the knickzones among the major rivers
and tributaries. At this point, only qualitative analysis for neighboring rivers is possible due to lack of
terrace age controls, and due to the sharp climatic gradient characteristic of the western North Qilian
Shan.

Please refer to our reply to the comments by Dr. Schwanghart, above, for further information. Our
detailed response for major river knickpoints is in line 25-47; our response for small river incisions is in
line 61-65; our response for the relationship between main stem incision and tributary knickpoints
formation in line 93-113; our plans to improve our manuscript in line 66-82.

*I would appreciate more clarity in the way tectonic drivers controlling the profile geometry are ruled out.*
*The first argument presented is that the current channel is carved in bedrock and not just an excavation*
*of the old alluvial fill. I do not see the connection between this argument and incision due to a period of*
*increased uplift. I would appreciate some clarity on this point. The second argument about incision rates*
*being higher as uplift rates is good but it would be far more convincing if you present it in terms of total*
*uplift since the abandonment of T1 versus total incision. This point would come across a lot better with*
*more visual support within some figure. Also, there's no figure that visualizes where the folding happens.*
*Figure 3 would also benefit from having the locations of faults (active and now inactive) on the map,*
*similar to Wang et al. 2020.*

Thanks for your suggestions to bolster our argument that knickzone-formation was not tectonically
driven. We will use these to frame a clearer argument in our revision. Specifically, we will:

1.  Include folded T2 terrace profile in figure 4a along with T1 and river profile of Beida River.
       2.  Add the location of active and inactive faults and the active fold axis into Figure 3a.
       3.  Add a field photo of the inactive fault covered by T1/T2 terrace fill in the supplement.
       4.  Rephrase our argument into two main points: first, field evidence doesn't support the presence of
           an active fault under the knickzone; second, the tectonic uplift rate of faulting and folding is much
smaller than the incision rate. We will include a comparison of the incision rate and uplift rate since
           the abandonment of T1 (24 kyr) as well.

*This might be more of a side note, but judging from Wang et al. 2020, the knickzone is located just above*
*a thrust fault that became recently inactive, correct? Maybe it should be pointed out that this fault has*
*been inactive for a time exceeding the age of the terraces studied here.*

Yes, the knickzone is located just above an inactive fault, which has been inactive at least since the
penultimate glacial period (Wang et al., 2020). We will add the fault location to our map, and include
photo evidence of this fault as we described above.

*Line 60: Please, highlight the Hexi Corridor in figure 1. I had to look up the name.*

Thank you for the suggestion. We will improve our figure quality in our revised manuscript.

*Line 89: Seems like either a word is missing or something else is wrong with the sentence.*

Sorry about the confusion. We will change the sentence to "Flights of terraces inset below both T1 and
T2 terrace treads, marking progressive degradation of the terrace fill and incision into underlying
bedrock."

*Line 118: What grain size was the measurement performed on?*

The grain size of the loess is generally <100µm, we will include this into our revised manuscript.

*Line 180-2: The Loess was dated to 3.2kyrs, therefore this is a minimum age and not the age of the*
*terrace, correct? And same would apply to the incision rates.*

This is correct. The age of 3.2 kyr from the loess cap is a minimum age, which suggests the duration of
the rapid, 2nd incision stage could be even shorter, and thus the incision rate even faster. Because we
explicitly state that we calculate the maximum duration and the minimum incision rate of the 2nd stage,
treating the loess-cap date as a minimum date for the underlying terrace deposits does not affect our
results.

However, one should consider 3.2 kyr can be a close approximate of the terrace age because for the
western part of the North Qilian Shan and the Hexi Corridor, loess was deposited continuously at least since mid-early Holocene (Küster et al., 2006). Our other samples, taken from the bottom of the loess
       cover on top of an older terrace tread just across the canyon give ages of 5.7 kyr and 6.5 kyr (Wang et
       al., 2020), which suggests that loess started depositing in this area no later than 6.5 to 5.7 kyr B.P.
       Therefore, we consider the loess on the low terrace was deposited immediately after the abandonment
       of the terrace.

In the revised manuscript, we will include a clarification of the loess age, the justification for treating this
       as a terrace age, and explain how even if treated as a minimum date, it does not affect our conclusions.

       *Line 192-3: This is weird. Why would the river cut through bedrock instead of the soft alluvium?*

       This is related to the period of low incision rate prior to 9.5 kyr B.P. During that time, the Beida River
meandered and widened its bedrock channel. Parts of the river reach, especially close to the outlet,
       eroded into and flowed on top of a bedrock strath cut laterally as the canyon widened. Once the pace of
       incision increased post 9.5 kyr B.P., base level fall at the mountain front led to isolation of the river
       course within this bedrock part of the canyon floor. Once entrenched into bedrock, the river could not
       access the more easily mobilized alluvial fill deposits of its former canyon bottom. We will improve our
description of this process in our revised manuscript.

       *Line 234-243: This sounds good to me, but I feel like the whole paragraph could be a lot shorter if there
       was somewhere a geologic map of the Beida and the neighboring rivers, with the steep segments
       highlighted. Could also be in the supplement.*

Thank you for the suggestion. We will include a geologic map of Beida and neighboring rivers in the
       supplement and make the argument more concise.

       *Figure 1: I have two points that are more of a suggestion. It would be good if the color map could be
       changed to something that is perceptually uniform (e.g. https://www.fabiocrameri.ch/colourmaps/), and
please add an underlying hillshade to improve visibility of the topography.*

       Thank you for the suggestion. We will improve our figure quality in our revised manuscript.

       *Figure 3: The label of Fig. 2C needs to be changed to 3C. I find it very confusing that the samples are
       indicated by alphabetical labels, because intuitively I would think that the letters indicate the locations of
the picture panels. Please, use numbers or something else instead. Also, please highlight the geologic
       contacts. I cannot see the contact between alluvial fill and bedrock in figure 3D.*

       We apologize for the confusion about the labels. We will change the labels for sample sites, and
       highlight the contact between alluvial fill and bedrock in figure 3d.

**Reply to Dr. Sean Gallen**

*(1) Figures: Many of the figures are too small and difficult to follow. Some figure additions are needed (e.g., terrace stratigraphy in key locations as in Wang et al. (2020) Figure 4b,c and additional river profiles in the region). I also have several specific suggestions that I think can help improve the presentation of the figures, as well as a couple of suggestions for additional figures. I include the specifics in my detailed line-by-line comments below.*

We apologize for the figure quality; we will make sure to replace these with higher quality versions in our revised manuscript.

*(2) Structure: I struggled quite a bit with the structure and flow of the manuscript. I think that part of this has to do with the legacy of revisions of a previous version of the study. The motivating observation is the observed knickzones in the river profiles, but the key data is the incision history from the terraces. A cleaner presentation of the results might focus on the terraces first and then discuss the characteristics of the river profile in the context of the incision rates and patterns. This could also serve to streamline the discussion because the terraces record the incision pulse, which presumably generated the knickzone due to variable substrate erodibility. Currently, from the results onward, the terrace and river profile discussions are mixed together, making a really interesting story challenging to follow. I think that this mixing of analyses and discussion makes it more difficult for the reader to clearly visualize the key points of the conceptual model for knickzone formation – climate-driven incision pulse causes downcutting in the alluvial fan downstream of the mountain front and an upstream change in substrate erodibility results in the formation of the knickzone. This is in the paper, but it could be presented more clearly and concisely.*

Thanks for your suggestion. We realize that our results and discussions of river profiles and terraces are mixed together and may benefit from some restructuring.

Therefore, we will rearrange the "Results" section as follows:

4.1. River profiles and river patches. In this section we will move the part about river patches in section 4.1 of the original manuscript, and also include descriptions of tributary profiles with figures into this section

4.2. Foreland and hinterland terrace development. In this section we will include contents of terraces that's in section 4.1 of the original manuscript, and contents of the strath exposure in section 4.3. We will also include descriptions of terrace substrate with figures for sample locations, and describe terrace continuity within our research area.

4.3 Incision rate estimation. We will keep this section the same as the original manuscript.

For the "Discussion" section we plan to make following changes:

5.1 Channel width and bedrock incision rate (originally 5.3)

5.2 Beida River knickzone formation and incision stages (originally 5.1). We will also add evidence from tributary profiles/knickzones to further strengthen our argument of knickzone formation.

5.3 Coupled incision model for knickzone formation (originally 5.2).

5.4 Climatic implications from rapid incision of Beida River. In this section, we will combine the contents of 5.4 and 5.5 in the original manuscript together. We will also include a brief discussion of the possible climate implication of the 9.5 kyr terrace.

5.5 In this section, we will discuss how and why the major rivers and adjacent small rivers draining the
western North Qilian Shan respond differently to the same mid-Holocene climate event.

*(3) Support for the preferred interpretation (and maybe some additional restructuring): I think the authors can rely more on the geochronology to support their favored interpretation. For example, the emplacement T1 fill terrace corresponds (roughly) with rapid cooling around the LGM and T2 the timing*
*of the preceding full interglacial. This information is not new but presented in Wang et al. (2020), so it is unclear to me why the apparent correlation between fill terrace deposition and climate isn't presented in the background section. Doing so would place the link between climate and incision-aggradation cycles in the reader's mind early and key them into thinking about temporal links between the terrace stratigraphy and climate as they read through the rest of the manuscript. Additionally, I think that the*
*authors can more strongly interpret the timing of inset terrace deposition with regional and global climate records and that this would help their arguments. For example, T1' is abandoned roughly at the Preboreal-Boreal transition. This seems like a missed opportunity in the discussion.*
*Furthermore, in looking at Google Earth, I think observations from adjacent drainages can be used to the benefit of the climate and discharge interpretations. Importantly, it is evident from Google Earth that the*
*only rivers that incise into the alluvial fan/fill north of the range front are those that drain high elevations. This is clearly recognized by the author's and mentioned in their interpretations, but what is missing are observations from satellite imagery (or Google Earth) and river profiles (not just from the trunk channels) from adjacent rivers to help support this kind of interpretation. This would be a small added figure/analysis but could help convince a skeptical reader.*

Thank you for the suggestion. We will include the T1 and T2 and their climate correlation in the background, and revisit this in the discussion.

We recognize that abandonment of T1' may correspond to a regional increase in incision rate near the end of the last glacial termination. This event seems to be recorded by several lakes located in the Northwest China (i.e., Hartmann, 2009; Jiang et al, 2008; Li et al, 2009; Rhodes et al, 1996; Wang et al,
2013); we will add this information to our revised manuscript.

As you point out, the incision history of adjacent drainages can provide additional evidence for the climate hypothesis we propose. We will focus our discussion of adjacent drainages on two main aspects:

1. Adjacent major rivers have similar deeply incised canyons and prominent knickzones. This supports that the transition from aggradation to incision of the foreland of these major rivers was a regional,
climatically driven event.

2. Adjacent small rivers and tributaries have only incised shallowly into the foreland-basin alluvial fill, which we interpret to be due to lack of rainfall and glacial coverage in lower elevations where these rivers originate. This is further evidence of how discharge strongly affects the incision behavior of rivers of the North Qilian Shan.

*(4) Slope Patch: I was a little confused by some aspects of the slope patch approach used here. I understand the formulation in Royden and Perron (2013), and I see what the authors are trying to do. My main question has to do with the observation of variable channel width that is not included in the analysis here. In the slope patch formulation in Royden and Perron (2013), they use the general stream power model (E = KA^mS^n). This equation presumably accounts for/assumes hydrologic (Q) and hydraulic (W) scalings with A encapsulated by exponent m. To handle these along stream changes, Royden and Perron (2013) use a coordinate transformation of distance, introducing the non-dimensional distance chi. From chi, along with other non-dimensionalization, Royden and Perron (2013) come up with their elegant slope patch solutions.*

*As I understand it for this application, Q is assumed to be uniform (or approximately so) over the length of the study area; however, W is shown to vary. It seems that equation 3 only works as applied in the manuscript if W is also uniform with only S and I varying along the channel length. It seems like the observed variations in W make it difficult to apply this simplified version of the slope patch approach as presented in equations 2-5.*

*That said, I do think that this analysis is useful, but I think that the caveats and assumptions made in this simple analysis need to be discussed and explained fully. As written, I think many readers might not realize that this analysis is limited to channel sections where Q and W are assumed to be uniform or that one needs independently constrained incision records for this kind of analysis. These requirements make the approach limiting and not general. I think these limitations/data requirements needed to be more clearly articulated. I also think the slope patch discussion could be rounded out by linking the modeled changes in incision rate and river profile geometry through time back to the observed slope patches in section 5.2. For example, how close are the observed and modeled slope patch gradients when run to the modern?*

*I realize this is a long comment that can probably be handled with the addition of a few sentences in the methods and discussion; I just wanted to make sure that I completely communicated my questions and tried to articulate where and why I was a little confused.*

The referee's comment highlights an inadvertent point of confusion where we describe our model approach. We do not require discharge to vary with time in our model, only that incision rate at base level varied over time, driving formation of the three slope patches we observe. Changes in Q are a likely cause of the increase of incision rate during formation of patch 2, but not required (e.g. a decrease of sediment flux may also explain enhanced incision of the foreland basin). Specifically, eq. 6 through 9 involve only incision rate and channel slope, not discharge or channel width directly. In response to the referee's comment, we will clarify in the methods section that our model does not strictly require changes in discharge with time. In the discussion of channel width and discharge variation that follows, we point out how both changes in channel width and discharge may explain aspects of the incision history, and that increased discharge overall, and especially during the second incision stage, are likely explanations for the observations (as would be a decrease in sediment flux).

We agree to referee that we need to emphasis this analysis is limited to channel sections where Q and W are assumed to be uniform. We will add this to our methods section. Our modeled results generally fit well with the observed slope patch. We will include a comparison of the modeled result with the observed river profile in our revision.

*(5) Terrace stratigraphic relationships: It is really hard to understand the general terrace stratigraphy without photos and figures. One suggestion is to add a composite terrace stratigraphy for several*
*locations along the river to illustrate the relationships between the fill terraces and the inset terraces. What I have in mind is something like figure 4b and c in Wang et al. (2020). It was hard for me to understand the context of the terrace observations without a figure like this. Also, from figure 4 in Wang et al. (2020), it looks like the T1' terrace is a cut terrace (meaning it is cut into the T1 fill terrace) and is not a strath terrace. Is this correct? If so, does it become a strath terrace up valley? Also, how might this*
*play into the interpretations forwarded in the study (i.e., strath vs. cut terraces)?*

Yes, T1' and inset terraces below T1' are all cut terraces. The 'strath' in our manuscript refers to the bedrock (paleo canyon floor and walls) below the terrace fills, we will clarify this in our revision. T1 and T1' are the terraces that can be traced and correlated along the river, which means they were formed while the river was more or less stable (or at a steady state). Terraces below T1' are local terraces that
cannot be correlated, formed while the river was incising and meandering as it did so. We will add this information and figures illustrating terrace relationships as we described in our reply for comment 2.

*Detailed line-by-line comments.*
*L 13: incision accelerated from what value to 25 m/ky?*
From 6 m/ky. We will rephrase the sentences to make it clearer.

*L 14: "faster" rather than "larger"?*
We will change to faster.

*L 17: "We interpret that this period of increased…"?*
We will make change to this sentence as suggested.

*L 28-29: The role of lithology is critical here and probably deserved more discussion in section 5 somewhere. Without the change in erodibility due to the lithology change at the range front, the*
*knickpoints would be causing the profile to relax (assuming elevated incision is due to an increase in Q, as suggested). It seems worth explaining this to the reader in some detail later.*

Thanks for the suggestion. We will emphasize the role of lithology in section 5.1.

*L 43 [Figure 1]: Make the map of figure 1 larger, label some of the key geographic features mentioned in*
*the text on the map (e.g., Hexi corridor), provide a zoomed-in shaded relief map of the location with the three highlighted rivers, and combine the current figure 1 and 2. Doing this would introduce the reader to all of the general key observations without having to flip back and forth and look in google earth to understand much of what is written. This might also be a good location to show some google earth images of the studied and adjacent rivers (see part of my major comment 3 above). Also, are the glaciers*
*in this map active or from the Pleistocene?*
Thank you for the suggestion, we will improve figure 1 as recommended. The glaciers in the map are active, we will add this information in the caption.

*L 51 [Figure 3]: The photos in figure 3 are very small but really important. Please make them bigger. It*
*would also be useful to add some photos showing the key relationship that the river didn't reoccupy the*
*same valley filled with T1.*
We will enlarge figure 3. We have already included multiple photos showing the key relationship that
the river didn't reoccupy the same valley filled with T1 in the supplement (Figure S3).

*L 66-72: Is there any information on Pleistocene ELAs or glacial extents in this region? It might be*
*relevant.*
The ELA during the LGM is ~400 m lower than present. We will include this information in our revised
manuscript.

*L 85-94: The typical convention in name/labeling terraces is that the older units have lower numbers and*
*the younger units have higher numbers. This is the same convention as bedrock map units. It appears*
*that in this study and in Wang et al. (2020) the opposite is used (older have higher numbers than*
*younger). Because of this, I found this section very confusing the first time I read it until I saw figure 5.*
*Also, why not label the inset terraces with something like letters? The lack of naming for them makes*
*reading a little awkward.*
More than one naming convention exists for flights of terraces. We have followed the labeling system
previous researchers have developed for the North Qilian Shan and other rivers within northwest China
so that it would be easier to cross reference with other publications. We hesitated to label inset terraces
below T1' in our original manuscript because they are local features abandoned as the river
progressively cut its gorge (e.g., Merritts et al., 1994) and therefore its difficult or impossible to correlate
other terraces upstream and downstream. The most extensive terraces present are labeled T2, T1, T1'.
For local inset terraces, we revise our manuscript to label the terraces below T1' as T1'a, T1'b, T1'c, etc.
in figures and descriptions; we will also emphasize that these labels do not imply that we can correlate
these along the river.

*L 99: I can't see the reverse fault offsets in figure 1.*
We will add the fault in figure 2a.

*L 114: inset rather than insect.*
Sorry about the mis-spelling. We will correct the typos and grammar mistakes in our manuscript.

*L 119 [Table 1]: Can a row be added to link these ages to the terraces they were collected from? Having a*
*composite terrace stratigraphy figure preceding this table and naming/labeling the inset terraces will*
*make this pretty easy to do.*
Yes, we will add a row to link the ages to the terraces.

L 124: see my major comment 4 regarding this section.
Please refer to our reply to comment 4.

L 128: Need to link A and Q in some kind of statement.
We realize that we do not use A in our calculations going forward, so we decided to remove A here,
because the term Q/W is sufficeint for our application of the stream power equation.

*L 136: km^2*
*L 137: km^2*

We apologize for our careless mistake. We will correct this in our revised manuscript.

*L 162-164: These changes in width impact hydraulic geometry and thus shear stress imparted by a flow of a given magnitude. How might these changes in channel width impact the assumptions made in the simplified slope patch calculations?*

We discussed this effect in the discussion section (5.3). Generally, narrowing of the channel leads to enhancement of incision rate. This is most obvious for patch 1 and patch 3, which have similar slopes but different incision rates (patch 3 is narrower and thus incises faster). We will briefly mention this effect in the revised manuscript.

*L 171-182: This section would be easier to follow if the inset terraces were labeled or named, and there were some figures showing their composite stratigraphy in a few key locations along the river.*

We will improve the introduction of the terraces, as we describe in reply for comment 2.

*L 184 [Figure 4]: can the y-axis label be oriented as in figure 2 with the labels reading from bottom to top? Also, I don't understand panel b, and I assume the y-axis is mislabeled (should be elevation rather than width). How does the channel bed have negative elevation?*

Panel b is mislabeled and should be height. We apologize for this mistake. The channel bed downstream of the mountain front is negative because we set the elevation of the channel bed at the mountain front

as 0 (base level). We will reorient the y-axis label.

*L195-198: Photos of these critical relationships would really help!*

We have included photos of exposure of the T1 strath in supplement (Figure S3).

*L 204 [Figure 5]: I don't find panel **a** very helpful because the fill terraces (besides T1) aren't the main part of this manuscript, but the inset terraces are. I can't see the distribution of the inset terraces here with respect to T1, so I don't get much out of this panel. Panel c should be enlarged to make it easier to read.*

We will highlight the inset terraces with another color, and we will enlarge panel c.

*L216-218: What about tectonic subsidence? No correction is needed for that?*

Thank you for pointing out that subsidence also contributes to the comparison of terrace elevations. Unlike for the hinterland, where we have good constraints on uplift from deformed terraces, we do not have a marker of subsidence of the foreland. However, in our 2020 paper, we did estimate tectonic

subsidence from our mechanical model of folding and found this to be a fraction (<10%) of the hangingwall uplift. The subsidence rate mainly affects the comparison of our 4.5 kyr and 3.2 kyr terraces, which bracket the most rapid period of incision. The 4.5 kyr terrace, located in the hangingwall, is corrected for ~4m of tectonic uplift. Correction of the 3.2 kyr terrace, located in the footwall, for tectonic subsidence would be less than 0.5 m. Rather than introduce a model for subsidence in this

paper, we will state that terraces in the foreland are not corrected for tectonic subsidence, because this is a small fraction of the hangingwall uplift, and cite our 2020 paper as a source for this information.

*L235-237: It would be nice to see some images of this from Google Earth or something. These observations are very helpful to the interpretations presented here.*

We will include field photographs of the incised gorges of these three rivers as a supplement figure.

*L239: Be careful here. The change in erodibility of the alluvial fan and the bedrock is essential to explain the knickpoint. The driver of incision might be a temporal increase in Q, but if that occurs in uniform substrate, the river profile will relax, forming an "inverted" knickpoint that migrates upstream. Here, the inferred elevated Q forces the river gradient to relax in the alluvial fan (incision), and this base level fall steepens the "harder" bedrock rock reach upstream. The interaction of the inferred climate change along with the spatial change in substrate erodibility explains the observations. Both are required.*

The change of lithology we mention here is the lithology underlying the present knickzone location. We will clarify this point so that it is clear that we are not referring to where the knickzone formed, at the mountain front.

*L245-249: Yes, changes in substrate erodibility are needed as stated here.*

*L305-324: It seems like this would be better placed before the slope patch discussion. It also makes me wonder about the utility/generality of the slope patch approach used here. The Royden and Perron (2013) formulation implicitly takes changes in width into account via the calculation of chi, here that doesn't work. It might be helpful to highlight this point earlier in section 3.3 and mention that this point will be discussed in detail in section 5.*

Similar to our reply for major comment 4, we treat width as constant within a patch, therefore the simplification of the chi function (eq. 3) is valid. We will place this section before the slope patch discussion, and make a clearer statement of the role of the channel width.

*L 325: It seems like showing the relaxation of the river profiles beneath the incised alluvial fan surface for a few of the rivers would help support the point that there is a reduction in the river channel gradient beyond the range front. In the framework used here, the two most obvious explanations are changes in water discharge and/or sediment flux. Considering that the rivers that show this behavior all drain high elevations, the interpretations forwarded here seem reasonable to me. The key observation that isn't explained well in my mind is that the river gradients have declined downstream of the knickzones in the alluvial fans. The gradient decline, coupled with the change in substrate erodibility, generates the knickzones, correct?*

Yes, that is our interpretation. In our manuscript we referred the gradient decline of river channels in the foreland as fast incision/base level drop since we are focusing on the changes at the mountain front. In our revision, we will point out more clearly in the background that the depth of canyon incision into the foreland decreases downstream. In our discussion, we will bring this observation into our justification for an increase in discharge or decrease of sediment flux as driver(s) of knickpoint formation.

*L 378-395: I am left wondering if it might be helpful to explain the possible links between climate and the previously dated T1 and T2 fill terraces earlier in the study. This is all based on published data and will help establish a link between climate and geomorphology in the study area before diving into the inset terraces. It can be revisited here, but much of this section is related to previous studies and knowledge gained, so it seems better suited for the background.*

We will include the T1 and T2 and their climate correlation in the background, and revisit this in the discussion.

**Reply to Dr. Chris Sheehan**

Major Comment:

*My only major comment concerns some of the assumptions that go into the coupled incision model and the discussion of its results. The model assumes a simplified Beida River incisional history occurring in three stages: Stage 1 (~9.5 – 4.7 ka, ~5.63 m/kyr), Stage 2 (~4.7 – 3.2 ka, ~18.6 m/kyr), and Stage 3 (~3.2 ka to present, ~11.56 m/kyr). Using the incision model, these discrete stages suggest that the knickzone was created over a duration of ~700 years with a minimum incision ate of ~50 m/kyr. The actual incision*
*history was likely much more complex, but I think that simplifying it into three stages for the sake of this analysis is perfectly justifiable.*

*However, I am curious to know how slightly different yet plausible incision scenarios might affect the model results. In particular, the boundary between Stages 2 and 3 is inferred to be 3.2 ka. However, this is only based on a single datapoint (a single OSL sample from sampling site B). One could imagine a more*
*complex incisional history between 4.7 ka (the boundary between Stages 1 and 2) and present. For example, perhaps the incision rate has gradually decreased since ~4.7 ka, implying that Stages 2 and 3 would display an exponential curve on Figure 7 rather than two linear segments. Alternatively, perhaps the terrace at site B was abandoned much later than implied by the ~3.2 ka depositional age, justifying the inclusion of a fourth Stage in the model.*

*I think the authors should add a few sentences in Section 5 that discuss how their model results might vary under different circumstances. Importantly, they should address how their conclusion that the knickzone was created over ~700 years at a minimum rate of ~50 m/kyr might vary. I don't think they need to perform a full sensitivity analysis (though that would be interesting!), nor do they need to quantify the duration and magnitude of knickzone formation under specific, alternative conditions.*
*Rather, I think they should just briefly list some plausible scenarios that could either increase or decrease the implied duration of knickzone formation and qualitatively discuss the effects. I think this could be done just by adding a few sentences to Section 5.1.*

*I listed this as a major comment because the 700-year duration and 50 m/kyr incision rate are a major takeaway of this study (I expect that they will immediately grab the readers' attention in the abstract).*
*Therefore, I think it is very important to put them in context.*

Thank you for bringing up this important point of understanding. The three-stage incision history we derive is supported by both the terrace record of channel incision and the existence of three distinct slope patches along the bedrock channel. We agree with the referee that the ages of the inset terraces inform, but do not define, the stage boundary ages. Incision may have been more complex in detail, but
major deviations from the three-stage model should have produced a different channel profile than what we observe. What is less certain in our model is the exact position, in time, of the stage boundaries, and therefore the maximum rate of incision during the second stage. We will make this point clearer in the conclusion of our revised manuscript. In response to other reviewer comments, we also plan to discuss further how age control affects our model uncertainty (e.g. that the 3.2 kyr age is
from a capping loess).

Minor Comments:

*1. I think that Table 1 could benefit from two changes. First, it is extremely difficult to tell which sample name the 1-sigma and 2-sigma results correspond to. I recommend realigning the data in the calibrated age column so that their corresponding sample names are unambiguous. Second, I recommend adding a column showing the elevational position of each sampling site relative to T1. This will allow the readers to quickly reference the relative age of each terrace.*

We will rearrange the table to make it more reader friendly, and adding a column to show the terrace height relative to T1.

*2. I have an issue with one of the authors' arguments against the knickzone being a lithologic feature. I do agree with the authors' interpretation that the knickzones on the Beida River and its neighboring rivers were created by climatic forcing and are not likely the result of lithologic variations (i.e. variations in K in equations 1-3). The strongest evidence ruling out a lithologic origin is the lack of a knickzone preserved on the T1 strath (lines 240-241) and the continuous projection of terraces across patch 2 (Figure 6).*

*However, consider this excerpt from Lines 236-239: "In the neighboring Maying and Hongshuiba River, similar to the Beida River, present river channels also incised into Late Pleistocene fill terraces..., forming prominent knickzones 10-15 km upstream of the mountain front (Figure 2). This suggests these knickzones share similar origins, reflecting regional forcing. Local variations of lithology would thus be an unlikely cause for knickzone formation." If I am interpreting this correctly, the authors argue that if the Beida River knickzone was a lithologic feature, we would not expect to see similar knickzones on the Maying or Hongshuiba rivers because the lithologic variations along the Beida River would not likely be present along the other two rivers. I disagree with this reasoning. I am unfamiliar with the regional geology of the North Qilian Shan, but I can see on Google Earth that there are SW-dipping lithologic contacts along the Beida River corridor. These contacts are roughly parallel along their NW-SE strike, and so without a more detailed geologic map, it seems entirely reasonable that the Beida, Maying, and Hongshuiba rivers might cross the same bedrock units in each of their knickzones. Also, it appears on Google Earth that the Beida River crosses lithologic contacts (marked by stark color contrasts) at the transition from patch 1-2 and patch 2-3.*

*I recommend that the authors examine the spatial relationships between the Beida River patches and the underlying lithologic contacts. I still agree with their interpretation that the knickzone is most likely a transient feature created by climatic forcing, but if it happens that the knickzone is underlain by a low-erodibility rock type, then this might be a contributing factor worthy of a brief discussion. Alternatively, if the boundaries between the three patches do not correspond with major lithologic transitions, then this will strengthen their interpretation. The authors could also extend this analysis to the other two rivers. Since I am unfamiliar with this region, I do not know if the authors can obtain a geologic map with enough detail to perform this analysis. If they cannot, I would be satisfied if they just removed their argument in Lines 236-239 and relied on their evidence in Lines 240-241.*

Thank you for your suggestion. We realize our lithology argument in Lines 236-239 of the manuscript is not strong enough. We will add a geologic map of the region with the knickzones of the three rivers highlighted, and include it as a figure in the supplement.

*3. Between Lines 363 to 371, it is unclear to me whether some sentences are information from Tan et al.,*
*2018 (cited in Line 362) or the authors' own interpretations. I think that these require clarification. Specifically, these lines:*

*• Lines 363- 364: "Between 24 kyr and 9.5 kyr B.P., the Beida River drainage was under the dominant influence of the arid westerly moisture source."*

This is our suggested scenario based on the climate records we presented in the paragraph above this
line. We realize we haven't clearly introduced this idea and need to cite the sources properly first.

*• Lines 365-367: "During the Early Holocene, the humid Asian monsoon expanded to the central North Qilian Shan, where it affected the Hei He main stem and filled Juyanze lake to its highest lake level."*

This is based on Herzschuh et al., 2004 and Hartmann and Wünnemann, 2009. We have mentioned this once in the background, but we should revisit this evidence again before we present our conclusion.

*• Lines 368-371: "During the mid-Holocene, the Asian monsoon grew stronger, starting in the Hei He drainage around 5.4 to 5.1 kyr B.P., and then expanding further to the western North Qilian Shan a few hundred years later. This peak of monsoon influence lasted less than 700±340 yr, which led to an increase of precipitation therefore an increase of water discharge and incision rate."*

"the Asian monsoon grew stronger, starting in the Hei He drainage around 5.4 to 5.1 kyr B.P" is based on
Herzschuh et al., 2004 and Hartmann and Wünnemann, 2009.

"then expanding further to the western North Qilian Shan a few hundred years later. This peak of monsoon influence lasted less than 700±340 yr, which led to an increase of precipitation therefore an increase of water discharge and incision rate." is our suggested scenario based on the lake records and our modeling results.

*If previous research demonstrated that the Asian monsoon expanded into the North Qillian Shan in the early Holocene and then strengthened during the mid Holocene, then this independently supports the authors' conclusions that the knickzone was created by mid Holocene climatic forcing. However, it is unclear to me whether this was previous research or a novel idea presented here.*

In our revision, we will concisely reintroduce these evidences in the discussion, then present our
conclusion.

*4. In Lines 335-346, the authors argue that a discharge contribution from melting glaciers would not have been enough to trigger knickzone formation. While this intuitively seems correct to me, I'm not sure that I agree with their reasoning. They cite metrics of modern glacial melt and discharge contributions,*
*compare these to possible conditions in the past, and conclude that mid Holocene melt contributions could not have triggered knickzone formation. However, the authors have not quantified the actual*

*discharge increase necessary to create the knickzone. Without this information, I don't think that the other metrics presented here definitively rule out glacial melt as the primary driving mechanism. I recommend modifying the sentence in Lines 345-346 to clarify this.*

We will revise our argument, and suggest the knickzone formation may be contributed by both the increase of precipitation and glacial melt.

*5. Can the authors comment on how the long-term (i.e. averaged over several glacial / interglacial cycles) bedrock incision rate may compare to the tectonic uplift rate? Beida River incision clearly*

*outpaces uplift during the transition from glacial to interglacial conditions, but averaged over $10^5$-year timescales, is the river more or less in steady-state? Quantifying this long-term trend requires dating several generations of strath terraces (see Pederson et al., 2006), and so this may not be possible to do for the Beida River. Still, it might be useful for the authors to add a few sentences discussing this in Section 5.*

Unfortunately, terrace straths older than T2 are largely absent from the Beida River canyon; removed by incision and widening of the canyon over time. The best, though less direct, information on long-term uplift rate is the exhumation rate measured from thermochronology (~1 m/kyr, i.e. Zheng et al., 2010). We will add a sentence citing these rates in the background of the revised manuscript, and cite these rates where we compare the Holocene incision rate to the tectonic uplift rate.

Line-by-line comments:
*• Line 114: "insect" should be "inset".*
*• Line 135: "drainage area of Beida River" should be "drainage area of the Beida River".*
*• Lines 136 and 137: "km2" should be "km$_2$".*
*• Lines 158, 245, and 384: "Beida River" should be "the Beida River".*
*• Lines 169 and 170: "above present riverbed" should be "above the present riverbed".*
*• Line 219: "Incision rate" should be "The incision rate".*
• Line 220: I think there may be unwanted spaces after some periods in this line. I think "between 4. 7 to 3. 2" should be "between 4.7 to 3.2".
*• Line 246: "It is likely that knickzone" should be "It is likely that the knickzone".*
*• Line 381: "correspond" should be "corresponding".*
*• Line 382: "dating of T2 terrace" should be "dating of the T2 terrace".*
*• Lines 382-384: This sentence is awkward. I recommend breaking it into two sentences and changing "similar to that T1" to "similar to how T1".*
*• Line 387: "similar age as T2 terrace" should be similar age as the T2 terrace".*
*• Line 388: "Shagou River" should be "the Shagou River".*

We will correct these typos and grammar mistakes in our revised manuscript.

*• Line 128: "A is drainage area" is unnecessary. Drainage area does not appear in this particular*
*formulation of the stream power equation (Q is used instead), and so the variable "A" does not need to be defined.*
Yes. We will remove A from our revision.

*• Lines 172-173: Is site C on the T1' terrace? If so, shouldn't the samples imply that T1' was abandoned AFTER 9.5 ± 0.16 ka (not prior)? The changes that I recommended for Table 1 could help clarify this.*

Yes, it should be after. We will correct this in our revision.

*• Lines 172-180: While reading this section, I found it difficult to construct a mental map of the terrace position / age relationship due to their heights being listed relative to different surfaces. This is later clarified very well in Figures 6 and 7, but those figures are neither shown nor referenced for another 5 pages. I think the changes I recommended for Table 1 will be very useful here, because the readers will have already seen these data, and they can flip back to the table for a visual aid if necessary.*

Thanks for the suggestion. We will make the changes to the table as recommended, and we will add a figure show the terrace structure for the sample site, and present a clearer description of the terraces in our revision.

*• Lines 198 199: Does this mean that none of the inset terraces below T1' have exposed straths?*

On the contrary, all inset terraces in the hinterland below T1 have well-exposed bedrock straths. Most of these were formed after the river had begun incising into bedrock. We will make sure to clarify this observation in our revision.

*• Line 229-230: Tying into my previous comment concerning long-term incision rates, over what period of time is the average uplift rate ~0.6?*

The ~0.6 m/kyr uplift rate is measured post the abandonment of T1 (24 kyr) terrace. We will clarify this in our revision.

**References**

Chen, R., Han, C., Liu, J., Yang, Y., Liu, Z., Wang, L. and Kang, E.: Maximum precipitation altitude on the
northern flank of the Qilian Mountains, northwest China, Hydrol. Res., 49(5), 1696–1710, doi:10.2166/nh.2018.121, 2018.

Crosby, B. T., Whipple, K. X., Gasparini, N. M. and Wobus, C. W.: Formation of fluvial hanging valleys: Theory and simulation, J. Geophys. Res. Earth Surf., 112(3), 1–20, doi:10.1029/2006JF000566, 2007.

Geng, H., Pan, B., Huang, B., Cao, B. and Gao, H.: The spatial distribution of precipitation and topography
in the Qilian Shan Mountains, northeastern Tibetan Plateau, Geomorphology, 297, 43–54, doi:10.1016/j.geomorph.2017.08.050, 2017.

Hartmann, K. and Wünnemann, B.: Hydrological changes and Holocene climate variations in NW China, inferred from lake sediments of Juyanze palaeolake by factor analyses, Quat. Int., 194(1–2), 28–44, doi:10.1016/j.quaint.2007.06.037, 2009.

Jiang, Q., Shen, J., Liu, X. and Zhang, E.: Holocene climate reconstructions of Ulungur Lake (Xinjiang, China) inferred from ostracod species assemblages and stable isotopes, Front. Earth Sci. China, 2(1), 31–40, doi:10.1007/s11707-008-0007-z, 2008.

Küster, Y., Hetzel, R., Krbetschek, M. and Tao, M.: Holocene loess sedimentation along the Qilian Shan (China): Significance for understanding the processes and timing of loess deposition, Quat. Sci. Rev.,
25(1–2), 114–125, doi:10.1016/j.quascirev.2005.03.003, 2006.

Li, Y., Wang, N., Cheng, H., Long, H. and Zhao, Q.: Holocene environmental change in the marginal area of the Asian monsoon: A record from Zhuye Lake, NW China, Boreas, 38(2), 349–361, doi:10.1111/j.1502-3885.2008.00063.x, 2009.

Merritts, D. J., Vincent, K. R. and Wohl, E. E.: Long river profiles, tectonism, and eustasy: a guide to
interpreting fluvial terraces, J. Geophys. Res., 99(B7), 14031–14050, doi:10.1029/94jb00857, 1994.

Rhodes, T. E., Gasse, F., Lin, R., Fontes, J. C., Wei, K., Bertrand, P., Gibert, E., Mélières, F., Piotr, T., Wang, Z. and Cheng, Z. Y.: A Late Pleistocene-Holocene lacustrine record from Lake Manas, Zunggar (northern Xinjiang, western China), Palaeogeogr. Palaeoclimatol. Palaeoecol., 120(1–2), 105–121, doi:10.1016/0031-0182(95)00037-2, 1996.

Wang, W., Feng, Z., Ran, M. and Zhang, C.: Holocene climate and vegetation changes inferred from pollen records of Lake Aibi, northern Xinjiang, China: A potential contribution to understanding of Holocene climate pattern in East-central Asia, Quat. Int., 311, 54–62, doi:10.1016/j.quaint.2013.07.034, 2013.

Zheng, D., Clark, M. K., Zhang, P., Zheng, W. and Farley, K. A.: Erosion, fault initiation and topographic
growth of the North Qilian Shan (northern Tibetan Plateau), Geosphere, 6(6), 937–941, doi:10.1130/GES00523.1, 2010.

---

## Author Response (AR1)

**Replies to comments**

**Reply to Dr. Wolfgang Schwanghart:**

*1. The three rivers shown in Fig. 2 have in common that they have a prominent knickzone as well as that they incise into the foreland-basin. The along-river distance of the knickzone to the mountain front is in all cases similar. According to the stream power incision model, however, slope patches should move upstream at a velocity dictated by upstream area. Hence, the three rivers should have similar upstream areas. However, looking at the map, the three sites seem to have very different areas. A way to address*
*the effect of variabe drainage areas is to calculate chi as a horizontal along-river distance. This might be very helpful because it would provide additional evidence for (or against) a common base level drop, if all three knickzones should have similar chi values measured from the mountain front.*

Thank you for pointing out the interesting relationship between the knickzones among the major rivers. We have tried using chi plots to analyze these rivers, but eventually chose to use elevation versus distance
plots for the following reasons. First, discharge is not solely correlated to the distribution of drainage areas within these catchments. Specifically, the annual precipitation is higher (>300 mm/yr) at high elevation, but very low (~100 mm/yr) at mountain front and foreland (Chen et al., 2018). In addition, tributaries at higher elevation (over 4000 m) are supplied continuously during the summer by glacier melt, while tributaries close to the mountain front are ephemeral streams with no glacial coverage. Second, there is
a strong east-west precipitation gradient for the North Qilian mountain range. For example, the annual precipitation at the mountain front decreases from 200-300 mm/yr in Maying River drainage to 100-200 mm/yr in Beida River drainage (Geng et al., 2017). The above evidence suggests that in the western North Qilian Shan, topographic and geographic factors strongly influence the discharge of each of the major drainages, which not only makes drainage area a less effective proxy of water discharge, but also makes
it difficult to compare different drainages by normalizing drainage areas.

In general, both slope and discharge contribute to knickzone retreat rate. Among the three major rivers draining the North Qilian Shan, each with incised foreland valleys and prominent knickpoints, the amount of incision varies by a factor of two. Specifically, at the mountain front, the height of the T1 terrace above the Maying River is ~250 m, whereas it is ~190m for the Hongshuiba River T1 and ~130 m for the Beida
River. These heights are controlled by the amount and rate of base level drop which is mainly controlled by the incision process within the foreland. The foreland incision rate depends on both changes in discharge and in sediment flux. For the Beida River, we have measurements of the incision rate from stream terraces available to us, but the other rivers lack such information. Possible reasons why the knickzone has retreated a similar distance on all three rivers thus include (1) differences in initial
knickpoint slope and the slopes of upstream/downstream patches, (2) higher precipitation in the east versus west, and (3) possibly an earlier onset time for the period of most rapid incision in the east versus the west. We included this into "Discussion", part of the section 5.4 (L458-471).

*2. It is interesting to note that some of the rivers draining the North Qilian Shan have steeply incised in*
*the foreland while others have not, at least those that seem to have smaller drainage areas (I have not investigated this in detail, though, so this may be a bit speculative). Looking at GEarth, it seems that in*

*particular those rivers with small catchments upstream the hanging wall are rather accumulating and form large fans with no (at least to me) obvious trend towards incision. Where these rivers drain into the gorges carved by the Beida river (and the other larger rivers), these smaller rivers are fluvial hanging*

*valleys and show signs of headward erosion. How is it possible that under a general increase in precipitation, there are such different patterns in river incision? Or are there other processes responsible for the incision? For example, could a lack of sediment connectivity and increased sediment storage behind terminal moraines in the higher areas drained by the major rivers be a possible explanation for sediment starvation of the main rivers and thus incision? At least, this explanation would not be at odds*

*with the data as well as your interpretation of the other terraces, as well.*

Regions at lower elevation and close to the mountain front receive the least precipitation, and have little or no glacier coverage to provide steady summer runoff. Similar to the downstream tributaries of the major rivers, small rivers draining the mountain front are ephemeral and dry for most of the time, even during the summer monsoon season. Therefore, we suggest the overall low water discharge to these low-elevation streams is the cause for the observed difference in incision pattern.

To address above concerns regarding incision of the major rivers (1) versus small streams and tributaries (2), we made following changes to improve our manuscript:

1.  We included more climate background of our study area. Especially the precipitation distribution in the western North Qilian Shan (L62-68).

2.  We included more details on the major rivers (drainage area, annual discharge, canyon depth, glacial coverage, etc.) in the background (L83-92, table 1).
3.  We added a paragraph in "Discussion", in which (L458-471) we discussed how drainage area is not a reliable proxy for water discharge in this region because of the distribution of rainfall and glacially modulated runoff characteristic of the North Qilian Shan, and how all these factors lead major rivers and small rivers to respond differently.

*3. As noted in the previous comment, I think that the paper could benefit from some more geomorphometric analysis. So far, the authors mainly looked at the trunk streams of the rivers, and focused this analysis on the Beida River. In order to generalize their findings, however, some additional*

*work is required that shows that the findings are consistent with the other drainage basins having these knickzones. One interesting observation that could shed additional light on these river systems is the analysis of the tributaries to these rivers. Looking at the DEM in Google Earth reveals numerous fluvial hanging valleys tributary to the river downstream of the knickzone while they are missing upstream of the knickzone. Finally, comparison with incision rates measured elsewhere would help to judge the*

*plausibility of the incision rates which are extremely high.*

Most of tributaries below the main stem knickzone form waterfalls where they join the main stem. This suggests when these tributary knickzones formed, the main stem was incising very fast so that the incision rate of the tributaries could not keep up with the main stem. This observation is consistent with terrace records and our model calculations. After the main stem knickzone swept through the downstream reach, the tributary knickzones may have not retreated upstream for several reasons. First, we argue that these ephemeral tributaries do not have enough stream power to incise fast into the bedrock over such a short time period (~4 kyr). Second, the extreme steepness of the hanging valleys and waterfalls prohibits the headward migration (i.e., Crosby et al. 2007).

Contrary to the referee's comment, tributaries of the Beida River above the main-stem knickzone do
form knickzones. However, these tributaries exhibit gentler slope where they join the main stem than tributaries downstream of the main-stem knickzone. We interpret that these upstream tributary knickzones reflect an ongoing process of adjustment related to the increase in incision rate of the main stem since 9.5 kyr. As shown by fig. 7 of the revised manuscript, incision rate along patch 1 increased at the beginning of stage 1 (9.5 kyr), and this rate may not have varied much from stages 2 to 3, as the
steep main-stem knickzone that formed during stage 2 has not yet propagated past these upstream tributary junctions.  The incision history we derive through a combination of terrace dating and modeling explains well this difference in tributary behavior upstream and downstream of the main-stem knickzone.

In our revised manuscript, we include example tributary profiles (6 tributaries downstream of the main-
stem knickzone, 6 tributaries upstream of the main-stem knickzone) in the "Results" section (section 4.1, L184-189), and include a discussion of the tributary profiles in the "Discussion" section. In section 5.2 (L355-364) we discuss how the tributary knickzone patterns fits with the overall incision history; in section 5.4 (L458-471) we discuss why these tributaries cannot keep up pace with rapid incision of the main stem downstream of its knickzone.

*4. A short comment on the supplements: The quality of the pictures in the supplements is relatively bad. Consider storing the pdf with images in higher resolution.*

*5. A short comment on the figures: I partly found it difficult to read these figures. Perhaps I am getting old, but some of these figures (in particular photos) are very small.*

We apologize for the quality of these figures. We replaced these figures with higher quality versions in our revised manuscript. Also, convert to pdf also seems to lower the figure quality, and we've tried to solve this problem, but if the quality of the pdf for the revised manuscript is still low, please contact us and we will provide the original figure.

**Reply to Dr. Richard Ott**

*My pain point is that a more thorough morphometric analysis would benefit the manuscript. The authors argue for a precipitation increase which should be regional. Therefore, it would be good to have chi-plots of the Beida and neighboring rivers, including tributaries, for the lower river section where the knickzones are located. Are the knickzones migrating up the tributaries, too? Are they at the same chi-distance compared to the trunk rivers? If, e.g the knickpoints do not manage migrate up the tributaries*
*as fast the trunk (in chi-space), this could be an indication that glacial melt is indeed the controlling factor, whereas an even increase in precipitation should affect all streams in the region in the same way.*

Thank you for pointing out the interesting relationship between the knickzones among the major rivers and tributaries. At this point, only qualitative analysis for neighboring rivers is possible due to lack of terrace age controls, and due to the sharp climatic gradient characteristic of the western North Qilian
Shan.

Please refer to our reply to the comments by Dr. Schwanghart, above, for further information. Our detailed response for major river knickpoints is in line 15-37; our response for small river incisions is in line 51-65; our response for the relationship between main stem incision and tributary knickpoints formation in line 76-99 of this document.

*I would appreciate more clarity in the way tectonic drivers controlling the profile geometry are ruled out. The first argument presented is that the current channel is carved in bedrock and not just an excavation of the old alluvial fill. I do not see the connection between this argument and incision due to a period of increased uplift. I would appreciate some clarity on this point. The second argument about incision rates*
*being higher as uplift rates is good but it would be far more convincing if you present it in terms of total uplift since the abandonment of T1 versus total incision. This point would come across a lot better with more visual support within some figure. Also, there's no figure that visualizes where the folding happens. Figure 3 would also benefit from having the locations of faults (active and now inactive) on the map, similar to Wang et al. 2020.*

Thanks for your suggestions to bolster our argument that knickzone-formation was not tectonically driven. We made the following changes to frame a clearer argument in our revision:

1. Included folded T2 terrace profile in figure 3a along with T1 and river profile of Beida River.
2. Added the location of active and inactive faults and the active fold axis into Figure 2a.
3. Added a field photo of the inactive fault covered by T1/T2 terrace fill in the supplement (Figure S6).
4. Rephrased our argument into two main points (L309-319): first, field evidence doesn't support the presence of an active fault under the knickzone; second, the tectonic uplift rate due to faulting and folding is much smaller than the incision rate. We also include a comparison of the incision rate and uplift rate since the abandonment of T1 (24 kyr), and compare these rates to independent exhumation rate information for the North Qilian Shan.

*This might be more of a side note, but judging from Wang et al. 2020, the knickzone is located just above a thrust fault that became recently inactive, correct? Maybe it should be pointed out that this fault has been inactive for a time exceeding the age of the terraces studied here.*

Yes, the knickzone is located just above an inactive fault, which has been inactive at least since the penultimate glacial period (Wang et al., 2020). We added the fault location to our map (Figure 2a), and included photo evidence of this fault as we described above (Figure S6).

*Line 60: Please, highlight the Hexi Corridor in figure 1. I had to look up the name.*

Thank you for the suggestion. We added Hexi Corridor in figure 1 in our revision.

*Line 89: Seems like either a word is missing or something else is wrong with the sentence.*

Sorry about the confusion. We changed the sentence to "Flights of terraces inset below both T1 and T2 terrace treads, marking progressive degradation of the terrace fill and incision into underlying bedrock."

*Line 118: What grain size was the measurement performed on?*

The grain size of the loess is generally <100μm, we included this into our revised manuscript (table 3 note).

*Line 180-2: The Loess was dated to 3.2kyrs, therefore this is a minimum age and not the age of the terrace, correct? And same would apply to the incision rates.*

This is correct. The age of 3.2 kyr from the loess cap is a minimum age, which suggests the duration of the rapid, 2nd incision stage could be even shorter, and thus the incision rate even faster. Because we explicitly state that we calculate the maximum duration and the minimum incision rate of the 2nd stage, treating the loess-cap date as a minimum date for the underlying terrace deposits does not affect our results.

However, one should consider 3.2 kyr can be a close approximation of the terrace age because for the western part of the North Qilian Shan and the Hexi Corridor, loess was deposited continuously at least since mid-early Holocene (Küster et al., 2006). Our other samples, taken from the bottom of the loess cover on top of an older terrace tread just across the canyon give ages of 5.7 kyr and 6.5 kyr (Wang et al., 2020), which suggests that loess started depositing in this area no later than 6.5 to 5.7 kyr B.P. Therefore, we consider the loess on the low terrace began to be deposited immediately after the abandonment of the terrace.

In the revised manuscript, we included a clarification of the loess age, the justification for treating this as a terrace age, and explained how even if treated as a minimum date, it does not affect our conclusions (L260-264; L393-398).

*Line 192-3: This is weird. Why would the river cut through bedrock instead of the soft alluvium?*

This is related to the period of relatively low incision rate prior to 9.5 kyr B.P. During that time, the Beida River meandered and widened its bedrock channel. Parts of the river reach, especially close to the outlet, eroded into and flowed on top of a bedrock strath cut laterally as the canyon widened. Once the pace of incision increased post 9.5 kyr B.P., base level fall at the mountain front led to isolation of the river course within this bedrock part of the canyon floor. Once entrenched into bedrock, the river could not access the more easily mobilized alluvial fill deposits of its former canyon bottom. We included this discussion in L338-342 in our revision.

*Line 234-243: This sounds good to me, but I feel like the whole paragraph could be a lot shorter if there was somewhere a geologic map of the Beida and the neighboring rivers, with the steep segments highlighted. Could also be in the supplement.*

Thank you for the suggestion. We included a geologic map of Beida and neighboring rivers in the
supplement (Figure S1) and made the argument more concise (L321-322).

*Figure 1: I have two points that are more of a suggestion. It would be good if the color map could be changed to something that is perceptually uniform (e.g. https://www.fabiocrameri.ch/colourmaps/), and please add an underlying hillshade to improve visibility of the topography.*

Thank you for the suggestion. We improved our figure quality in our revised manuscript as suggested.

*Figure 3: The label of Fig. 2C needs to be changed to 3C. I find it very confusing that the samples are indicated by alphabetical labels, because intuitively I would think that the letters indicate the locations of the picture panels. Please, use numbers or something else instead. Also, please highlight the geologic*
*contacts. I cannot see the contact between alluvial fill and bedrock in figure 3D.*

We apologize for the confusion about the labels. We changed the labels for sample sites into B-01 to B-05, and highlight the contact between alluvial fill and bedrock in figure 3d.

**Reply to Dr. Sean Gallen**

*(1) Figures: Many of the figures are too small and difficult to follow. Some figure additions are needed (e.g., terrace stratigraphy in key locations as in Wang et al. (2020) Figure 4b,c and additional river profiles in the region). I also have several specific suggestions that I think can help improve the presentation of the figures, as well as a couple of suggestions for additional figures. I include the specifics*
*in my detailed line-by-line comments below.*
We apologize for the figure quality; we replaced these with higher quality versions in our revised manuscript, and we added additional figures as suggested, specified in replies below

*(2) Structure: I struggled quite a bit with the structure and flow of the manuscript. I think that part of this*
*has to do with the legacy of revisions of a previous version of the study. The motivating observation is the observed knickzones in the river profiles, but the key data is the incision history from the terraces. A cleaner presentation of the results might focus on the terraces first and then discuss the characteristics of the river profile in the context of the incision rates and patterns. This could also serve to streamline the discussion because the terraces record the incision pulse, which presumably generated the knickzone due*
*to variable substrate erodibility. Currently, from the results onward, the terrace and river profile discussions are mixed together, making a really interesting story challenging to follow. I think that this mixing of analyses and discussion makes it more difficult for the reader to clearly visualize the key points of the conceptual model for knickzone formation – climate-driven incision pulse causes downcutting in the alluvial fan downstream of the mountain front and an upstream change in substrate erodibility*
*results in the formation of the knickzone. This is in the paper, but it could be presented more clearly and concisely.*

Thanks for your suggestion. We realize that our result and discussion of river profiles and terraces are mixed together and may benefit from some restructuring.

We rearranged the "Results" section as follows:

4.1. Beida River profiles. In this section we included the part about river patches in section 4.1 of the original manuscript, and also included descriptions of tributary profiles with figures into this section.

4.2. Beida River terrace development. In this section we included contents of terraces that's in section 4.1 of the original manuscript, and contents of the strath exposure originally in section 4.3. We also included descriptions of terrace substrate with figures for sample locations, and describe terrace
continuity within our research area.

4.3 Incision rate estimation. We kept this section the same as the original manuscript.

For the "Discussion" section we made following changes:

5.1 Channel width and bedrock incision rate (originally 5.3)

5.2 Beida River knickzone formation and incision stages (originally 5.1). We added evidence from
tributary profiles/knickzones to further strengthen our argument of knickzone formation; we also clarified the description for the different incision phases and stages.

5.3 Coupled incision model for knickzone formation (originally 5.2).

5.4 Climate implications of the Beida River incision history. In this section, we combined the contents of sections 5.4 and 5.5 in the original manuscript. We rearranged this section to discuss the climate implication for different incision stages. We also discussed how and why the major rivers and adjacent small rivers draining the western North Qilian Shan responded differently to the same mid-Holocene climate event.

*(3) Support for the preferred interpretation (and maybe some additional restructuring): I think the authors can rely more on the geochronology to support their favored interpretation. For example, the emplacement T1 fill terrace corresponds (roughly) with rapid cooling around the LGM and T2 the timing of the preceding full interglacial. This information is not new but presented in Wang et al. (2020), so it is unclear to me why the apparent correlation between fill terrace deposition and climate isn't presented in the background section. Doing so would place the link between climate and incision-aggradation cycles in the reader's mind early and key them into thinking about temporal links between the terrace stratigraphy and climate as they read through the rest of the manuscript. Additionally, I think that the authors can more strongly interpret the timing of inset terrace deposition with regional and global climate records and that this would help their arguments. For example, T1' is abandoned roughly at the Preboreal-Boreal transition. This seems like a missed opportunity in the discussion. Furthermore, in looking at Google Earth, I think observations from adjacent drainages can be used to the benefit of the climate and discharge interpretations. Importantly, it is evident from Google Earth that the only rivers that incise into the alluvial fan/fill north of the range front are those that drain high elevations. This is clearly recognized by the author's and mentioned in their interpretations, but what is missing are observations from satellite imagery (or Google Earth) and river profiles (not just from the trunk channels) from adjacent rivers to help support this kind of interpretation. This would be a small added figure/analysis but could help convince a skeptical reader.*

Thank you for the suggestion. We included the T1 and T2 and their climate correlation in the background (L96-98), and revisited this in the discussion (L410-415).

We recognize that abandonment of T1' may correspond to a regional increase in incision rate near the end of the last glacial termination. This event seems to be recorded by several lakes located in the Northwest China (i.e., Hartmann, 2009; Jiang et al, 2008; Li et al, 2009; Rhodes et al, 1996; Wang et al, 2013); we have added this information to our revised manuscript (L420-425).

As you point out, the incision history of adjacent drainages can provide additional evidence for the climate hypothesis we propose. We focused our discussion of adjacent drainages on two main aspects in our revision:

1. Adjacent major rivers have similar deeply incised canyons and prominent knickzones. This supports that the transition from aggradation to incision of the foreland of these major rivers was a regional, climatically driven event (L426-428).

2. Adjacent small rivers and tributaries have only incised shallowly into the foreland-basin alluvial fill, which we interpret to be due to lack of rainfall and glacial coverage in lower elevations where these rivers originate. This is further evidence of how discharge strongly affects the incision behavior of rivers of the North Qilian Shan (L458-472).

*(4) Slope Patch: I was a little confused by some aspects of the slope patch approach used here. I understand the formulation in Royden and Perron (2013), and I see what the authors are trying to do. My main question has to do with the observation of variable channel width that is not included in the analysis here. In the slope patch formulation in Royden and Perron (2013), they use the general stream power model (E = KA^mS^n). This equation presumably accounts for/assumes hydrologic (Q) and hydraulic (W) scalings with A encapsulated by exponent m. To handle these along stream changes, Royden and Perron (2013) use a coordinate transformation of distance, introducing the non-dimensional distance chi. From chi, along with other non-dimensionalization, Royden and Perron (2013) come up with their elegant slope patch solutions.*

*As I understand it for this application, Q is assumed to be uniform (or approximately so) over the length of the study area; however, W is shown to vary. It seems that equation 3 only works as applied in the manuscript if W is also uniform with only S and I varying along the channel length. It seems like the observed variations in W make it difficult to apply this simplified version of the slope patch approach as presented in equations 2-5.*

*That said, I do think that this analysis is useful, but I think that the caveats and assumptions made in this simple analysis need to be discussed and explained fully. As written, I think many readers might not realize that this analysis is limited to channel sections where Q and W are assumed to be uniform or that one needs independently constrained incision records for this kind of analysis. These requirements make the approach limiting and not general. I think these limitations/data requirements needed to be more clearly articulated. I also think the slope patch discussion could be rounded out by linking the modeled changes in incision rate and river profile geometry through time back to the observed slope patches in section 5.2. For example, how close are the observed and modeled slope patch gradients when run to the modern?*

*I realize this is a long comment that can probably be handled with the addition of a few sentences in the methods and discussion; I just wanted to make sure that I completely communicated my questions and tried to articulate where and why I was a little confused.*

The referee's comment highlights an inadvertent point of confusion where we describe our model approach. We do not require discharge to vary with time in our model, only that incision rate at base level varied over time, driving formation of the three slope patches we observe. Changes in Q are a likely cause of the increase of incision rate during formation of patch 2, but not required (e.g. a decrease of sediment flux may also explain enhanced incision of the foreland basin). Specifically, eq. 6 through 9 involve only incision rate and channel slope, not discharge or channel width directly.

In response to the referee's comment, we clarified in the methods section that our model does not strictly require changes in discharge with time (L163-165). In the discussion of channel width and discharge variation that follows, we point out how both changes in channel width and discharge may explain aspects of the incision history, and that increased discharge overall, and especially during the second incision stage, are likely explanations for the observations (as would be a decrease in sediment flux; L286-290).

We agree to the referee that we need to emphasis this analysis is limited to channel sections where Q and W are assumed to be uniform. We have added this to our methods section (L170-171).

Our modeled results generally fit well with the observed slope patch. We included a comparison of the modeled result with the observed river profile in our revision (L399-403, figure 8).

*(5) Terrace stratigraphic relationships: It is really hard to understand the general terrace stratigraphy without photos and figures. One suggestion is to add a composite terrace stratigraphy for several locations along the river to illustrate the relationships between the fill terraces and the inset terraces. What I have in mind is something like figure 4b and c in Wang et al. (2020). It was hard for me to*

*understand the context of the terrace observations without a figure like this. Also, from figure 4 in Wang et al. (2020), it looks like the T1' terrace is a cut terrace (meaning it is cut into the T1 fill terrace) and is not a strath terrace. Is this correct? If so, does it become a strath terrace up valley? Also, how might this play into the interpretations forwarded in the study (i.e., strath vs. cut terraces)?*

Yes, T1' and inset terraces below T1' are all cut terraces. The 'strath' in our manuscript refers to the bedrock (paleo canyon floor and walls) below the terrace fills, we will clarify this in our revision. T1 and T1' are the terraces that can be traced and correlated along the river, which means they were formed while the river was more or less stable (or at a steady state). Terraces below T1' are local terraces that cannot be correlated, formed while the river was incising and meandering as it did so.

We added this information into section 4.2, and clarified the description of the terrace relationships. We also added figures illustrating terrace relationships, as we also describe in our reply for comment 2.

*Detailed line-by-line comments.*
*L 13: incision accelerated from what value to 25 m/ky?*
From 6 m/ky. We rephrased the sentences to "over a duration of less than 1.5 kyr, during which incision accelerated from 6 m/kyr to at least 25 m/kyr.".

*L 14: "faster" rather than "larger"?*
We changed to faster.

*L 17: "We interpret that this period of increased…"?*
We changed this sentence as suggested.

*L 28-29: The role of lithology is critical here and probably deserved more discussion in section 5 somewhere. Without the change in erodibility due to the lithology change at the range front, the*

*knickpoints would be causing the profile to relax (assuming elevated incision is due to an increase in Q, as suggested). It seems worth explaining this to the reader in some detail later.*

Thanks for the suggestion. We emphasize the role of lithology in the revised discussion (L302-308).

*L 43 [Figure 1]: Make the map of figure 1 larger, label some of the key geographic features mentioned in*

*the text on the map (e.g., Hexi corridor), provide a zoomed-in shaded relief map of the location with the three highlighted rivers, and combine the current figure 1 and 2. Doing this would introduce the reader to all of the general key observations without having to flip back and forth and look in google earth to understand much of what is written. This might also be a good location to show some google earth images of the studied and adjacent rivers (see part of my major comment 3 above). Also, are the glaciers*

*in this map active or from the Pleistocene?*

Thank you for the suggestion, we improved figure 1 as recommended. The glaciers in the map are active, we added this information in the caption.

*L 51 [Figure 3]: The photos in figure 3 are very small but really important. Please make them bigger. It would also be useful to add some photos showing the key relationship that the river didn't reoccupy the same valley filled with T1.*
We enlarged figure 3. We have already included multiple photos showing the key relationship that the river didn't reoccupy the same valley filled with T1 in the supplement (Figure S5).

*L 66-72: Is there any information on Pleistocene ELAs or glacial extents in this region? It might be relevant.*
The ELA during the LGM is ~400 m lower than present. We included this information in our revised manuscript (L74-76).

*L 85-94: The typical convention in name/labeling terraces is that the older units have lower numbers and the younger units have higher numbers. This is the same convention as bedrock map units. It appears that in this study and in Wang et al. (2020) the opposite is used (older have higher numbers than younger). Because of this, I found this section very confusing the first time I read it until I saw figure 5. Also, why not label the inset terraces with something like letters? The lack of naming for them makes reading a little awkward.*
More than one naming convention exists for flights of terraces. We have followed the labeling system previous researchers have developed for the North Qilian Shan and other rivers within northwest China so that it would be easier to cross reference with other publications. We hesitated to label inset terraces below T1' in our original manuscript because they are local features abandoned as the river progressively cut its gorge (e.g., Merritts et al., 1994) and therefore its difficult or impossible to correlate other terraces upstream and downstream. The most extensive terraces present are labeled T2, T1, T1'. For local inset terraces, we revised our manuscript to label the terraces below T1' as T1'a, T1'b, T1'c, etc. in figures and descriptions (Figure 2, table 2); we also emphasized that these labels do not imply that we can correlate these along the river.

*L 99: I can't see the reverse fault offsets in figure 1.*
We added the fault in figure 2a.

*L 114: inset rather than insect.*
Sorry about the mis-spelling. We corrected the typos and grammar mistakes in our manuscript.

*L 119 [Table 1]: Can a row be added to link these ages to the terraces they were collected from? Having a composite terrace stratigraphy figure preceding this table and naming/labeling the inset terraces will make this pretty easy to do.*
Yes, we added a row to link the ages to the terraces (Now table 2).

L 124: see my major comment 4 regarding this section.
Please refer to our reply to comment 4.

L 128: Need to link A and Q in some kind of statement.
We realize that we do not use A in our calculations going forward, so we decided to remove A here, because the term Q/W is sufficient for our application of the stream power equation.

*L 136: km^2*
*L 137: km^2*

We apologize for our careless mistake. We corrected this in our revised manuscript.

*L 162-164: These changes in width impact hydraulic geometry and thus shear stress imparted by a flow*
*of a given magnitude. How might these changes in channel width impact the assumptions made in the*
*simplified slope patch calculations?*
We discuss this effect in the discussion section (5.3, now 5.1). Generally, narrowing of the channel leads
to enhancement of incision rate. This is most obvious for patch 1 and patch 3, which have similar slopes
but different incision rates (patch 3 is narrower and thus incises faster). Please refer to our reply to
comment 4 for more details.

*L 171-182: This section would be easier to follow if the inset terraces were labeled or named, and there*
*were some figures showing their composite stratigraphy in a few key locations along the river.*
We improved the introduction of the terraces in the revision, as we described in reply for comment 2.

*L 184 [Figure 4]: can the y-axis label be oriented as in figure 2 with the labels reading from bottom to*
*top? Also, I don't understand panel b, and I assume the y-axis is mislabeled (should be elevation rather*
*than width). How does the channel bed have negative elevation?*
Panel b is mislabeled and should be height. We apologize for this mistake. The channel bed downstream
of the mountain front is negative because we set the elevation of the channel bed at the mountain front
as 0 (base level). We fixed these mistakes in our revision

*L195-198: Photos of these critical relationships would really help!*
We have included photos of exposure of the T1 strath in supplement (Figure S5).

*L 204 [Figure 5]: I don't find panel **a** very helpful because the fill terraces (besides T1) aren't the main*
*part of this manuscript, but the inset terraces are. I can't see the distribution of the inset terraces here*
*with respect to T1, so I don't get much out of this panel. Panel c should be enlarged to make it easier to*
*read.*
We highlighted the inset terraces with another color, and we enlarged panel c in the revision.

*L216-218: What about tectonic subsidence? No correction is needed for that?*
Thank you for pointing out that subsidence also contributes to the comparison of terrace elevations.
Unlike for the hinterland, where we have good constraints on uplift from deformed terraces, we do not
have a marker of subsidence of the foreland. However, in our 2020 paper, we did estimate tectonic
subsidence from our mechanical model of folding and found this to be a fraction (<10%) of the
hangingwall uplift. The subsidence rate mainly affects the comparison of our 4.5 kyr and 3.2 kyr
terraces, which bracket the most rapid period of incision. The 4.5 kyr terrace, located in the hangingwall,
is corrected for ~4m of tectonic uplift. Correction of the 3.2 kyr terrace, located in the footwall, for
tectonic subsidence would be less than 0.5 m.
Rather than introduce a model for subsidence in this paper, we stated in the revised manuscript that
terraces in the foreland are not corrected for tectonic subsidence, because this is a small fraction of the
hangingwall uplift, and cited our 2020 paper as a source for this information (L252-255).

*L235-237: It would be nice to see some images of this from Google Earth or something. These observations are very helpful to the interpretations presented here.*

We included field photographs of the incised gorges of these three rivers as a supplement figure (Figure S2).

*L239: Be careful here. The change in erodibility of the alluvial fan and the bedrock is essential to explain the knickpoint. The driver of incision might be a temporal increase in Q, but if that occurs in uniform substrate, the river profile will relax, forming an "inverted" knickpoint that migrates upstream. Here, the inferred elevated Q forces the river gradient to relax in the alluvial fan (incision), and this base level fall steepens the "harder" bedrock rock reach upstream. The interaction of the inferred climate change along with the spatial change in substrate erodibility explains the observations. Both are required.*

The change of lithology we mention here is the lithology underlying the present knickzone location. We rephrased this in our revision as "A second alternative hypothesis is that the Beida River knickzone formed at its present location" (L319-320).

*L245-249: Yes, changes in substrate erodibility are needed as stated here.*

*L305-324: It seems like this would be better placed before the slope patch discussion. It also makes me wonder about the utility/generality of the slope patch approach used here. The Royden and Perron (2013) formulation implicitly takes changes in width into account via the calculation of chi, here that doesn't work. It might be helpful to highlight this point earlier in section 3.3 and mention that this point will be discussed in detail in section 5.*

Similar to our reply for major comment 4, we treat width as constant within a patch, therefore the simplification of the chi function (eq. 3) is valid. We placed this section before the slope patch discussion, and made a clearer statement of the role of the channel width (L285-290).

*L 325: It seems like showing the relaxation of the river profiles beneath the incised alluvial fan surface for a few of the rivers would help support the point that there is a reduction in the river channel gradient beyond the range front. In the framework used here, the two most obvious explanations are changes in water discharge and/or sediment flux. Considering that the rivers that show this behavior all drain high elevations, the interpretations forwarded here seem reasonable to me. The key observation that isn't explained well in my mind is that the river gradients have declined downstream of the knickzones in the*

*alluvial fans. The gradient decline, coupled with the change in substrate erodibility, generates the knickzones, correct?*

Yes, that is our interpretation. In our manuscript we referred the gradient decline of river channels in the foreland as fast incision/base level drop since we are focusing on the changes at the mountain front. In our revision, we point out in the background that the depth of canyon incision into the foreland decreases downstream (L90-92). In the "Results" section, we compare the present channel slope with the alluvial fan slope (L181-183). In our revised discussion, we bring this observation into our justification for an increase in discharge or decrease of sediment flux as driver(s) of knickpoint formation (L303-308).

*L 378-395: I am left wondering if it might be helpful to explain the possible links between climate and the previously dated T1 and T2 fill terraces earlier in the study. This is all based on published data and will help establish a link between climate and geomorphology in the study area before diving into the inset terraces. It can be revisited here, but much of this section is related to previous studies and knowledge gained, so it seems better suited for the background.*

We included the T1 and T2 and their climate correlation in the background (L97-98), and revisited this in the discussion (L410-415).

**Reply to Dr. Chris Sheehan**

Major Comment:

*My only major comment concerns some of the assumptions that go into the coupled incision model and the discussion of its results. The model assumes a simplified Beida River incisional history occurring in three stages: Stage 1 (~9.5 – 4.7 ka, ~5.63 m/kyr), Stage 2 (~4.7 – 3.2 ka, ~18.6 m/kyr), and Stage 3 (~3.2 ka to present, ~11.56 m/kyr). Using the incision model, these discrete stages suggest that the knickzone was created over a duration of ~700 years with a minimum incision ate of ~50 m/kyr. The actual incision history was likely much more complex, but I think that simplifying it into three stages for the sake of this analysis is perfectly justifiable.*

*However, I am curious to know how slightly different yet plausible incision scenarios might affect the model results. In particular, the boundary between Stages 2 and 3 is inferred to be 3.2 ka. However, this is only based on a single datapoint (a single OSL sample from sampling site B). One could imagine a more complex incisional history between 4.7 ka (the boundary between Stages 1 and 2) and present. For example, perhaps the incision rate has gradually decreased since ~4.7 ka, implying that Stages 2 and 3 would display an exponential curve on Figure 7 rather than two linear segments. Alternatively, perhaps the terrace at site B was abandoned much later than implied by the ~3.2 ka depositional age, justifying the inclusion of a fourth Stage in the model.*

*I think the authors should add a few sentences in Section 5 that discuss how their model results might vary under different circumstances. Importantly, they should address how their conclusion that the knickzone was created over ~700 years at a minimum rate of ~50 m/kyr might vary. I don't think they need to perform a full sensitivity analysis (though that would be interesting!), nor do they need to quantify the duration and magnitude of knickzone formation under specific, alternative conditions. Rather, I think they should just briefly list some plausible scenarios that could either increase or decrease the implied duration of knickzone formation and qualitatively discuss the effects. I think this could be done just by adding a few sentences to Section 5.1.*

*I listed this as a major comment because the 700-year duration and 50 m/kyr incision rate are a major takeaway of this study (I expect that they will immediately grab the readers' attention in the abstract). Therefore, I think it is very important to put them in context.*

Thank you for bringing up this important point of understanding. The three-stage incision history we derive is supported by both the terrace record of channel incision and the existence of three distinct slope patches along the bedrock channel. We agree with the referee that the ages of the inset terraces inform, but do not define, the stage boundary ages. Incision may have been more complex in detail, but major deviations from the three-stage model should have produced a different channel profile than what we observe. What is less certain in our model is the exact position, in time, of the stage boundaries, and therefore the maximum rate of incision during the second stage. We made this point clearer in the conclusion of our revised manuscript (L396-398). In response to other reviewer comments, we discuss further how age control affects our model uncertainty (e.g. that the 3.2 kyr age is from a capping loess) (L393-395). We also included a validation of the modeled results, which shows that our model provides a good explanation for the observed Beida River profile and incision history (L399-403; figure 8).

Minor Comments:

*1. I think that Table 1 could benefit from two changes. First, it is extremely difficult to tell which sample*

*name the 1-sigma and 2-sigma results correspond to. I recommend realigning the data in the calibrated age column so that their corresponding sample names are unambiguous. Second, I recommend adding a column showing the elevational position of each sampling site relative to T1. This will allow the readers to quickly reference the relative age of each terrace.*

We rearranged the table to make it more reader friendly, and adding a column to show the terrace height relative to T1.

*2. I have an issue with one of the authors' arguments against the knickzone being a lithologic feature. I do agree with the authors' interpretation that the knickzones on the Beida River and its neighboring rivers were created by climatic forcing and are not likely the result of lithologic variations (i.e. variations*

*in K in equations 1-3). The strongest evidence ruling out a lithologic origin is the lack of a knickzone preserved on the T1 strath (lines 240-241) and the continuous projection of terraces across patch 2 (Figure 6).*

*However, consider this excerpt from Lines 236-239: "In the neighboring Maying and Hongshuiba River, similar to the Beida River, present river channels also incised into Late Pleistocene fill terraces..., forming*

*prominent knickzones 10-15 km upstream of the mountain front (Figure 2). This suggests these knickzones share similar origins, reflecting regional forcing. Local variations of lithology would thus be an unlikely cause for knickzone formation." If I am interpreting this correctly, the authors argue that if the Beida River knickzone was a lithologic feature, we would not expect to see similar knickzones on the Maying or Hongshuiba rivers because the lithologic variations along the Beida River would not likely be*

*present along the other two rivers. I disagree with this reasoning. I am unfamiliar with the regional geology of the North Qilian Shan, but I can see on Google Earth that there are SW-dipping lithologic contacts along the Beida River corridor. These contacts are roughly parallel along their NW-SE strike, and so without a more detailed geologic map, it seems entirely reasonable that the Beida, Maying, and Hongshuiba rivers might cross the same bedrock units in each of their knickzones. Also, it appears on*

*Google Earth that the Beida River crosses lithologic contacts (marked by stark color contrasts) at the transition from patch 1-2 and patch 2-3.*

*I recommend that the authors examine the spatial relationships between the Beida River patches and the underlying lithologic contacts. I still agree with their interpretation that the knickzone is most likely a transient feature created by climatic forcing, but if it happens that the knickzone is underlain by a low-erodibility rock type, then this might be a contributing factor worthy of a brief discussion. Alternatively, if the boundaries between the three patches do not correspond with major lithologic transitions, then this will strengthen their interpretation. The authors could also extend this analysis to the other two rivers. Since I am unfamiliar with this region, I do not know if the authors can obtain a geologic map with enough detail to perform this analysis. If they cannot, I would be satisfied if they just removed their argument in Lines 236-239 and relied on their evidence in Lines 240-241.*

Thank you for your suggestion. We realize our lithology argument in Lines 236-239 of the manuscript is not strong enough. We added a geologic map of the region with the knickzones of the three rivers highlighted, and included it as a figure in the supplement (Figure S1). We also clarified our discussion of how lithology and faulting do not appear to have let to initiation of the knickzone at its present location.

*3. Between Lines 363 to 371, it is unclear to me whether some sentences are information from Tan et al., 2018 (cited in Line 362) or the authors' own interpretations. I think that these require clarification. Specifically, these lines:*

*• Lines 363- 364: "Between 24 kyr and 9.5 kyr B.P., the Beida River drainage was under the dominant influence of the arid westerly moisture source."*

This is our suggested scenario based on the climate records we presented in the paragraph above this line. We realize we haven't clearly introduced this idea and needed to cite the sources properly first in the revised manuscript.

*• Lines 365-367: "During the Early Holocene, the humid Asian monsoon expanded to the central North Qilian Shan, where it affected the Hei He main stem and filled Juyanze lake to its highest lake level."*

This is based on Herzschuh et al., 2004 and Hartmann and Wünnemann, 2009. We have mentioned this once in the background, but we now revisit this evidence again before we present our conclusion.

*• Lines 368-371: "During the mid-Holocene, the Asian monsoon grew stronger, starting in the Hei He drainage around 5.4 to 5.1 kyr B.P., and then expanding further to the western North Qilian Shan a few hundred years later. This peak of monsoon influence lasted less than 700±340 yr, which led to an increase of precipitation therefore an increase of water discharge and incision rate."*

"the Asian monsoon grew stronger, starting in the Hei He drainage around 5.4 to 5.1 kyr B.P" is based on Herzschuh et al., 2004 and Hartmann and Wünnemann, 2009.

"then expanding further to the western North Qilian Shan a few hundred years later. This peak of monsoon influence lasted less than 700±340 yr, which led to an increase of precipitation therefore an increase of water discharge and incision rate." is our suggested scenario based on the lake records and our modeling results. We have clarified these points in the revision.

*If previous research demonstrated that the Asian monsoon expanded into the North Qillian Shan in the early Holocene and then strengthened during the mid Holocene, then this independently supports the authors' conclusions that the knickzone was created by mid Holocene climatic forcing. However, it is unclear to me whether this was previous research or a novel idea presented here.*

In our revision, we rearranged our discussion to concisely reintroduce the evidence for a mid-Holocene strengthening of the monsoon before presenting our conclusion (L430-459).

*4. In Lines 335-346, the authors argue that a discharge contribution from melting glaciers would not have been enough to trigger knickzone formation. While this intuitively seems correct to me, I'm not sure that I agree with their reasoning. They cite metrics of modern glacial melt and discharge contributions, compare these to possible conditions in the past, and conclude that mid Holocene melt contributions could not have triggered knickzone formation. However, the authors have not quantified the actual discharge increase necessary to create the knickzone. Without this information, I don't think that the other metrics presented here definitively rule out glacial melt as the primary driving mechanism. I recommend modifying the sentence in Lines 345-346 to clarify this.*

We revised our argument and suggest that increased glacial melt may also have contributed to knickzone formation (L427-430).

*5. Can the authors comment on how the long-term (i.e. averaged over several glacial / interglacial cycles) bedrock incision rate may compare to the tectonic uplift rate? Beida River incision clearly outpaces uplift during the transition from glacial to interglacial conditions, but averaged over $10^5$-year timescales, is the river more or less in steady-state? Quantifying this long-term trend requires dating several generations of strath terraces (see Pederson et al., 2006), and so this may not be possible to do for the Beida River. Still, it might be useful for the authors to add a few sentences discussing this in Section 5.*

Unfortunately, terrace straths older than T2 are largely absent from the Beida River canyon; removed by incision and widening of the canyon over time. The best, though less direct, information on long-term uplift rate is the exhumation rate measured from thermochronology (~1 m/kyr, i.e. Zheng et al., 2010). We added a sentence citing these rates in the background of the revised manuscript (L58-59), and cited these rates where we compare the Holocene incision rate to the tectonic uplift rate (L315-316).

Line-by-line comments:
• *Line 114: "insect" should be "inset".*
• *Line 135: "drainage area of Beida River" should be "drainage area of the Beida River".*
• *Lines 136 and 137: "km2" should be "km$_2$".*
• *Lines 158, 245, and 384: "Beida River" should be "the Beida River".*
• *Lines 169 and 170: "above present riverbed" should be "above the present riverbed".*
• Line 219: "Incision rate" should be "The incision rate".
• Line 220: I think there may be unwanted spaces after some periods in this line. I think "between 4. 7 to 3. 2" should be "between 4.7 to 3.2".

*• Line 246: "It is likely that knickzone" should be "It is likely that the knickzone".*
*• Line 381: "correspond" should be "corresponding".*
*• Line 382: "dating of T2 terrace" should be "dating of the T2 terrace".*
*• Lines 382-384: This sentence is awkward. I recommend breaking it into two sentences and changing "similar to that T1" to "similar to how T1".*
*• Line 387: "similar age as T2 terrace" should be similar age as the T2 terrace".*
*• Line 388: "Shagou River" should be "the Shagou River".*

We corrected these typos and grammar mistakes in our revised manuscript.

*• Line 128: "A is drainage area" is unnecessary. Drainage area does not appear in this particular formulation of the stream power equation (Q is used instead), and so the variable "A" does not need to be defined.*

Yes. We removed A from our revision.

*• Lines 172-173: Is site C on the T1' terrace? If so, shouldn't the samples imply that T1' was abandoned AFTER 9.5 ± 0.16 ka (not prior)? The changes that I recommended for Table 1 could help clarify this.*

Yes, it should be after. We corrected this in our revision.

*• Lines 172-180: While reading this section, I found it difficult to construct a mental map of the terrace position / age relationship due to their heights being listed relative to different surfaces. This is later clarified very well in Figures 6 and 7, but those figures are neither shown nor referenced for another 5*

*pages. I think the changes I recommended for Table 1 will be very useful here, because the readers will have already seen these data, and they can flip back to the table for a visual aid if necessary.*
Thanks for the suggestion. We made the changes to the table as recommended, and we added a figure showing the terrace structure for the sample site, and presented a clearer description of the terraces in our revision (section 5.2).

*• Lines 198 199: Does this mean that none of the inset terraces below T1' have exposed straths?*
All the inset terraces are cut terraces into T1 fill. We clarified this in our revision (L219).

*• Line 229-230: Tying into my previous comment concerning long-term incision rates, over what period of*

*time is the average uplift rate ~0.6?*
The ~0.6 m/kyr uplift rate is measured post the abandonment of T1 (24 kyr) terrace. We clarified this in our revision as: "Based on our previous research (Wang et al., 2020), the average vertical uplift rate at the mountain front since the abandonment of T1 is only ~0.6 m/kyr."

**References**

Chen, R., Han, C., Liu, J., Yang, Y., Liu, Z., Wang, L. and Kang, E.: Maximum precipitation altitude on the northern flank of the Qilian Mountains, northwest China, Hydrol. Res., 49(5), 1696–1710, doi:10.2166/nh.2018.121, 2018.

Crosby, B. T., Whipple, K. X., Gasparini, N. M. and Wobus, C. W.: Formation of fluvial hanging valleys: Theory and simulation, J. Geophys. Res. Earth Surf., 112(3), 1–20, doi:10.1029/2006JF000566, 2007.

Geng, H., Pan, B., Huang, B., Cao, B. and Gao, H.: The spatial distribution of precipitation and topography in the Qilian Shan Mountains, northeastern Tibetan Plateau, Geomorphology, 297, 43–54, doi:10.1016/j.geomorph.2017.08.050, 2017.

Hartmann, K. and Wünnemann, B.: Hydrological changes and Holocene climate variations in NW China, inferred from lake sediments of Juyanze palaeolake by factor analyses, Quat. Int., 194(1–2), 28–44, doi:10.1016/j.quaint.2007.06.037, 2009.

Jiang, Q., Shen, J., Liu, X. and Zhang, E.: Holocene climate reconstructions of Ulungur Lake (Xinjiang, China) inferred from ostracod species assemblages and stable isotopes, Front. Earth Sci. China, 2(1), 31–40, doi:10.1007/s11707-008-0007-z, 2008.

Küster, Y., Hetzel, R., Krbetschek, M. and Tao, M.: Holocene loess sedimentation along the Qilian Shan (China): Significance for understanding the processes and timing of loess deposition, Quat. Sci. Rev., 25(1–2), 114–125, doi:10.1016/j.quascirev.2005.03.003, 2006.

Li, Y., Wang, N., Cheng, H., Long, H. and Zhao, Q.: Holocene environmental change in the marginal area of the Asian monsoon: A record from Zhuye Lake, NW China, Boreas, 38(2), 349–361, doi:10.1111/j.1502-3885.2008.00063.x, 2009.

Merritts, D. J., Vincent, K. R. and Wohl, E. E.: Long river profiles, tectonism, and eustasy: a guide to interpreting fluvial terraces, J. Geophys. Res., 99(B7), 14031–14050, doi:10.1029/94jb00857, 1994.

Rhodes, T. E., Gasse, F., Lin, R., Fontes, J. C., Wei, K., Bertrand, P., Gibert, E., Mélières, F., Piotr, T., Wang, Z. and Cheng, Z. Y.: A Late Pleistocene-Holocene lacustrine record from Lake Manas, Zunggar (northern Xinjiang, western China), Palaeogeogr. Palaeoclimatol. Palaeoecol., 120(1–2), 105–121, doi:10.1016/0031-0182(95)00037-2, 1996.

Wang, W., Feng, Z., Ran, M. and Zhang, C.: Holocene climate and vegetation changes inferred from pollen records of Lake Aibi, northern Xinjiang, China: A potential contribution to understanding of Holocene climate pattern in East-central Asia, Quat. Int., 311, 54–62, doi:10.1016/j.quaint.2013.07.034, 2013.

Zheng, D., Clark, M. K., Zhang, P., Zheng, W. and Farley, K. A.: Erosion, fault initiation and topographic growth of the North Qilian Shan (northern Tibetan Plateau), Geosphere, 6(6), 937–941, doi:10.1130/GES00523.1, 2010.

---

## Referee Report (RR1)

Dear Editor and Corresponding Authors,

It was a pleasure reading the revised manuscript from Wang et al. The manuscript has been greatly improved from its previous version, and all my comments have been adequately addressed. I believe that the current version of the manuscript is nearly ready for publication. I have only three very minor formatting comments, none of which I believe warrant a new draft to be circulated before final acceptance.

In my previous letter, I echoed the suggestion to include chi plots of the three major rivers. In their response to Dr. Schwanghart's first comment, the authors explain that drainage area is not an effective proxy for discharge in this field area due to strong vertical and longitudinal precipitation gradients (I agree with this point). While they don't explicitly say so, the authors seem to imply that when they performed a chi analysis, the knickzones did not have similar chi values, and they attribute this to the poor correlation between drainage area and discharge (this would make sense to me). If this is indeed the case, I think it might still be useful to include the chi plots and discuss them in Lines 443-471 (dissimilar chi values due to a poor drainage area / discharge correlation is itself an interesting result potentially worthy of discussion). That being said, I think the manuscript is still suitable for publication without them, and so I will defer to the authors and the other referees on the matter.

Once again, if the authors have any specific questions regarding my comments, please feel free to contact me.

Chris Sheehan
Boston College Department of Earth and Environmental Sciences
sheehacz@bc.edu
* * *
Minor Comments:

Line 177: "(Figure 3c." should be "(Figure 3c).". There is a missing closed-parenthesis.
Line 465: "fast" should be "faster".
Figure 4a: Please add an elevation scale / color bar.

---

## Author Response (AR2)

Reply to Dr. Richard Ott:

*"The authors did a good job at addressing most of my initial comments. The discussion on tectonic and lithologic effects for knickpoint formation have been improved and easier to follow. I really appreciate the new figure S6, highlighting that the reverse fault at the Beida knickzone has been inactive since T2 formation. The figure quality in general has improved. I feel like this manuscript can be published after addressing some minor comments.*

*I understand the reservations towards calculating chi with a constant drainage area-discharge relationship. However, the observation of different knickpoint locations along the major rivers is still very interesting, and deserves a more quantitative analysis. The authors point out regional precipitation gradients that have been constrained in other studies. I suggest to use these data to calculate precipitation corrected chi-values and see if this correction makes the knickpoints line up in chi-space. If there are not enough data to calculate precipitation corrected chi for all the drainage basins, the authors should at least qualitatively discuss if the known precipitation gradients would bias the knickpoints along the main stems into the right direction (e.g. less knickpoint migration in streams with lower precipitation)."*

Thank you for the suggestion. We have included a supplement figure (figure S7), which shows the chi values and adjusted chi values of the three rivers. The chi plot normalizing with upstream drainage area, shows that rivers in the east retreat more than rivers in the west. We suggest this east-west difference is partially caused by the east-west precipitation gradient. We therefore adjusted the chi plot by normalizing based on the present discharge at the mountain front of the three rivers. This adjusted chi plot brings the normalized knickzone locations of Maying and Hongshuiba River closer to the Beida River knickzone location, while the eastward trend of increasing normalized retreating distance still exist. We suggest both the timing and amount of precipitation during the mid-Holocene pluvial period could account for the remaining difference. We suggest these evidences enhanced our argument of monsoon and orographic effects strongly impacts the incisional behavior of the rivers. We have incorporated this interpretation into the Discussion section, in lines 459-484.

*L57: typo. Northeastern.*
*L59: There is a typo in the exhumation rate. I assume the unit is meant to be 1m/kyr.*

*L465: Typo.*
We apologize for these mistakes. We have corrected the typos.

*L89-91: The authors use the word "canyons", but mean the incision into the Pleistocene alluvial fans. The word canyon could also apply to the river valleys in the mountain range. Please rephrase.*

We rephrased the sentence as: "The canyons of the three rivers are deepest at the mountain front where the North Qilian fault juxtaposes bedrock against Quaternary sediments, ~130 m, ~190 m, and ~240 m for Beida, Hongshuiba, and Maying River, respectively"

*L126: Typo. I think the authors meant IntCal13. (by the way, an updated calibration curve has been published, IntCal 20).*

We corrected it to "IntCal13". Thank you for bring our attention to the newest calibration curve updates. We have checked the calibration results of IntCal20 and compared with the old ages. For all the samples, the differences between the two calibration approaches are generally less than 10 yr, and won't affect our interpretation and modelling of the incision history. For this reason, and also as an attempt to keep our age data internally consistent, we decided to keep the IntCal13 results as it is.

Reply to Dr. Chris Sheehan:

*It was a pleasure reading the revised manuscript from Wang et al. The manuscript has been greatly*
50 *improved from its previous version, and all my comments have been adequately addressed. I believe that the current version of the manuscript is nearly ready for publication. I have only three very minor formatting comments, none of which I believe warrant a new draft to be circulated before final acceptance. In my previous letter, I echoed the suggestion to include chi plots of the three major rivers. In their response to Dr. Schwanghart's first comment, the authors explain that drainage area is*
55 *not an effective proxy for discharge in this field area due to strong vertical and longitudinal precipitation gradients (I agree with this point). While they don't explicitly say so, the authors seem to imply that when they performed a chi analysis, the knickzones did not have similar chi values, and they attribute this to the poor correlation between drainage area and discharge (this would make sense to me). If this is indeed the case, I think it might still be useful to include the chi plots and*
60 *discuss them in Lines 443-471 (dissimilar chi values due to a poor drainage area / discharge correlation is itself an interesting result potentially worthy of discussion). That being said, I think the manuscript is still suitable for publication without them, and so I will defer to the authors and the other referees on the matter.*

Thank you for your positive feedbacks on our revised manuscript. We agree that including the chi
65 analysis can be an enhancement of the argument that we put in the last part of the discussion. We therefore include chi plots based on drainage area and chi plots adjusted using discharge to the supplement, and rewrote lines 458-471. Details are in our reply to Dr. Richard Ott (lines 17-27).

*Line 177: "(Figure 3c." should be "(Figure 3c).". There is a missing closed-parenthesis.*

70 *Line 465: "fast" should be "faster".*

We apologize for these mistakes. We have corrected the typos.

*Figure 4a: Please add an elevation scale / color bar*

We have updated the figure and added the elevation scale.

75

80

85    Other minor revisions:

Line 5 (marked document): add affiliation.

Line 11, 92, 111, 410, 467: minor changes in elevation and distance values to keep these values consistent throughout the manuscript.

Line 46, 198, 389: add supplementing information.

90    Line 105: round decimals.

Typos and grammar corrections.